# Fire360: A Benchmark for Robust Perception and Episodic Memory in Degraded 360° Firefighting Video

**Aditi Tiwari[1], Farzaneh Masoud[2], Dac Trong Nguyen[2], Jill Kraft[2], Heng Ji[1], Klara Nahrstedt[1]**
[1]University of Illinois Urbana-Champaign    [2]Illinois Fire Service Institute
{aditit5, fmasoud2, dacn, jillks, hengji, klara}@illinois.edu

## Abstract

Modern AI systems struggle most in environments where reliability is critical-scenes with smoke, poor visibility, and structural deformation. Each year, tens of thousands of firefighters are injured on duty, often due to breakdowns in situational perception [35]. We introduce **Fire360**, a benchmark for evaluating perception and reasoning in safety-critical firefighting scenarios. The dataset includes 228 360° videos from professional training sessions under diverse conditions (e.g., low light, thermal distortion), annotated with action segments, object locations, and degradation metadata. Fire360 supports five tasks: Visual Question Answering, Temporal Action Captioning, Object Localization, Safety-Critical Reasoning, and **Transformed Object Retrieval** (TOR). TOR tests whether models can match pristine exemplars to fire-damaged counterparts in unpaired scenes, evaluating episodic memory under irreversible visual transformations. While human experts achieve 83.5% on TOR, models like GPT-4o lag significantly, exposing failures in reasoning under degradation. By releasing Fire360 and its evaluation suite, we aim to advance models that not only see, but also remember, reason, and act under uncertainty. *The dataset is available at https://uofi.box.com/v/fire360dataset.*

## 1 Introduction

Can AI save lives when smoke blinds even the bravest first responders? Firefighters operate amid dense smoke, collapsing structures, and thermal distortion-conditions where perception failures carry life-threatening consequences. In 2023 alone, U.S. firefighters sustained 63,175 injuries, with 65,650 recorded the year prior [35]. These high-risk environments demand not only robust perception, but also procedural awareness, temporal reasoning, and resilience to visual degradation. Yet current AI systems-especially those trained solely on text or synthetic imagery-lack the grounding needed to operate in such physically chaotic environments. Achieving human-level reliability requires understanding the real world-and that understanding is hard.

Human responders rely on procedural memory and causal reasoning to locate tools, assess hazards, and identify charred equipment under limited visibility. Vision-language models (VLMs), however, depend on intact features and degrade under occlusion or distortion. Existing research focuses on simple scenes and isolated objects, with three critical limitations: it relies on clean imagery unsuitable for low-visibility emergencies [21], uses synthetic simulations that lack real-world complexity [58], and overlooks the temporal reasoning and aggregation of knowledge needed to track object change [18].

However, even accurate perception does not guarantee operational understanding. **Transformation-invariant episodic memory** represents a critical capability gap in safety-critical environments. Firefighters routinely recognize degraded equipment such as melted helmets, charred hoses, or smoke-obscured tools by maintaining object identity across irreversible physical transformations. Current AI systems lack this capacity, failing once visual features deform beyond recognition. This motivates

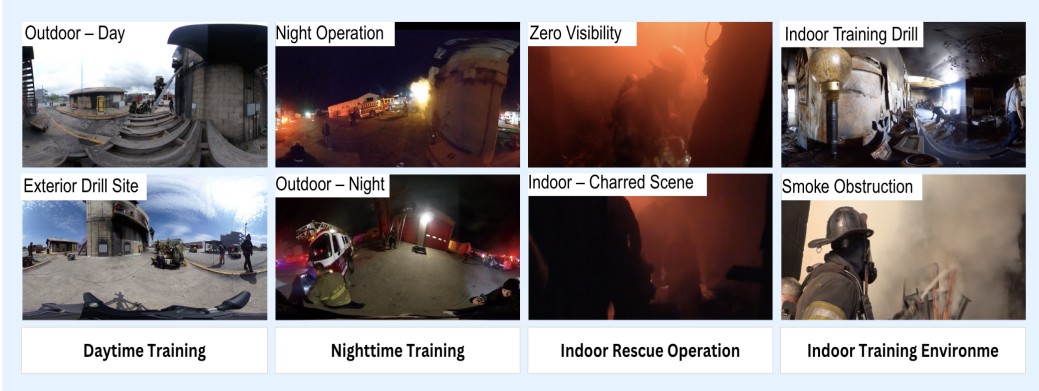

Figure 1: Example frames from Fire360, showcasing diverse operational settings and environmental conditions: (top row) outdoor firefighting scenes in day and night conditions, (bottom row) indoor low-visibility environments with dense smoke and limited lighting.

our **Transformed Object Retrieval (TOR)** task (Figure 6), which evaluates whether models can retrieve degraded counterparts of pristine exemplars without spatial or temporal continuity.

To address these gaps, we introduce **Fire360**, a benchmark built from 228 professionally recorded 360° firefighter training videos captured across day/night and indoor/outdoor conditions with high visual degradation. Each video follows nationally standardized drills [34], with certified instructors verifying annotations for actions, objects, and environments under realistic operational settings. Fire360 supports five tasks that probe distinct model competencies: Visual Question Answering (spatial reasoning), Temporal Action Captioning (temporal grounding), Object Localization (degradation robustness), Safety-Critical Reasoning (procedural compliance), and Transformed Object Retrieval (TOR), which evaluates recognition under irreversible object deformation.

In TOR, models must match a clean object exemplar to its fire-damaged counterpart-melted, occluded, or deformed-in an *unpaired* post-fire scene, meaning the reference and retrieval images come from different scenes with no temporal or spatial continuity, testing memory, material reasoning, and spatial grounding. Human experts achieve 83.5% accuracy; GPT-4o scores 39.8%. In VQA, Qwen-VL and LLaVA-1.5 reach 47.2% and 50.3% accuracy, while human performance remains at 91.4%. We benchmark CLIP (ViT-B/32) [41], BLIP-2 (OPT-6.7B) [25], GLaMM-7B [5], Grounding DINO (v1) [29], GPT-4o [37], Qwen-VL [2], and LLaVA-v1.5-13B [28], and observe consistent degradation-induced failures across tasks. By releasing Fire360's dataset, annotations, and toolkit, we lay the foundation for models that reason, remember, and act reliably in real-world high-risk operational environments.

Our main contributions can be summarized as follows:

- We introduce **Fire360**, a large-scale benchmark built from 228 professionally recorded 360° firefighter training videos captured under diverse real-world conditions (e.g., smoke, blur, low light), with certified expert-verified annotations.

- We define five evaluation tasks to isolate key reasoning failures in VLMs under environmental degradation.

- We propose **TOR**, a novel retrieval task requiring models to match pristine object exemplars to their fire-damaged counterparts in unpaired scenes. TOR evaluates transformation-invariant recognition and exposes a 43.7% performance gap between humans and state-of-the-art models, revealing fundamental failures in maintaining object identity under irreversible physical degradation.

## 2 Related Work

Fire360 intersects several active areas of research, including safety-critical AI systems, panoramic video understanding, robustness under degradation, and episodic memory in vision. We summarize the most relevant directions.

Table 1: Comparison with publicly available video datasets. ✓: Available, ✗: Not available.

| Dataset | Third-Person | 360° | Egocentric | Video | Audio | Real-world | Safety-Critical | Duration (s) | Public |
|---|---|---|---|---|---|---|---|---|---|
| Ego4D [16] | ✗ | ✗ | ✓ | ✓ | ✓ | ✓ | ✗ | 10,800,000 | ✓ |
| EPIC-Kitchens [9] | ✗ | ✗ | ✓ | ✓ | ✗ | ✓ | ✗ | 712,800 | ✓ |
| 360+x [4] | ✓ | ✓ | ✓ | ✓ | ✓ | ✗ | ✗ | 244,800 | ✓ |
| HACS++ [63] | ✓ | ✗ | ✗ | ✓ | ✗ | ✗ | ✗ | 500,400 | ✓ |
| **Fire360 (Ours)** | ✓ | ✓ | ✓ | ✓ | ✓ | ✓ | ✓ | **180,000** | ✓ |

**AI for Firefighting and Safety-Critical Domains.** Simulated environments such as FLAIM [13] and VR-based systems [44, 42, 58, 52] approximate fire scenarios using synthetic assets. Datasets like ACT360 [49] (55 videos, not public) introduce 360° action detection but omit protocol modeling, while FASDD [55], DFS [57], D-Fire [53], and Landsat-based systems [10, 12, 64] focus on fire classification without human-agent interaction. Fire360 addresses this gap by capturing real firefighting procedures, safety violations, and degradation effects in annotated 360° video. Recent panoramic datasets from NIST [36] further motivate the need for operational benchmarks grounded in real-world conditions.

**Panoramic and Egocentric Video Understanding.** Datasets like 360-Indoor [7], KITTI-360 [26], and 360+x [4] explore panoramic tasks in static or low-risk settings. Egocentric datasets such as Ego4D [16], EPIC-Kitchens [9], and EgoZAR [40] focus on manipulation and routine task recognition. Fire360 differs by targeting high-stakes, degraded environments and supporting tasks like Safety Reasoning and Transformed Object Retrieval. VIEW360 [47] similarly leverages panoramic views but is limited to anomaly detection in accessibility contexts. Table 1 contextualizes Fire360 within this landscape, highlighting our unique combination of 360° capture, safety-critical focus, and systematic degradation modeling.

**Robust Perception and Memory-Augmented Vision.** Recent surveys and methods highlight adversarial training, sparse attacks, and spatio-temporal augmentation as key techniques for robust video perception [50, 33, 61]. Fire360 contributes by offering real-world degradation (e.g., smoke, occlusion) rather than synthetic perturbations. Memory-augmented models [51, 30, 56] provide frameworks for episodic retrieval and reasoning, which Fire360 operationalizes in safety-critical scenarios via the TOR task.

**Transformation-Aware Retrieval and World Modeling.** Fire360 builds on retrieval datasets addressing object state change [43, 22, 23] but uniquely tests recovery under irreversible physical degradation. TOR requires models to maintain identity across scenes without continuity, aligning with recent interest in learned world models and episodic prediction [18, 45, 11]. Section 5 introduces our Transformed Object Retrieval (TOR) task, which addresses these limitations by requiring retrieval across unpaired, fire-transformed scenes where objects undergo irreversible material degradation.

## 3 The Fire360 Dataset

**Overview.** Fire360 contains 228 professionally recorded emergency response videos totaling 180,000 seconds (50 hours), captured at one of the oldest firefighter training institutes in North America under strict safety and privacy protocols. Each recording documents real drills conducted under standardized national procedures and supervised by 3 to 6 certified instructors. All footage excludes personal identifiers and focuses solely on team-based operational scenarios. Video was captured using a Ricoh Theta V 360° camera at 3840×1920 resolution and 60 frames per second, with spherical imagery internally stitched into equirectangular panoramas. For outdoor scenes, the 360° camera was tripod-mounted at average human eye level (∼5 feet). For indoor environments where the tripod could not withstand high temperatures, the camera was either helmet-mounted or handheld by firefighters, resulting in egocentric or wearable-view perspectives. Temporally aligned audio is included in the dataset but is not used in this release.

To support a range of modeling assumptions, Fire360 also provides rectilinear renderings that approximate the field-of-view of conventional 2D cameras. These projections facilitate research into spatial grounding, distortion-aware perception, and compatibility with pipelines that do not natively support 360° formats. Some videos are stored in 2D format to highlight complex firefighting actions in zoomed detail, but retain metadata for reconstruction into 360° views. Figure 2 illustrates the dual-view support.

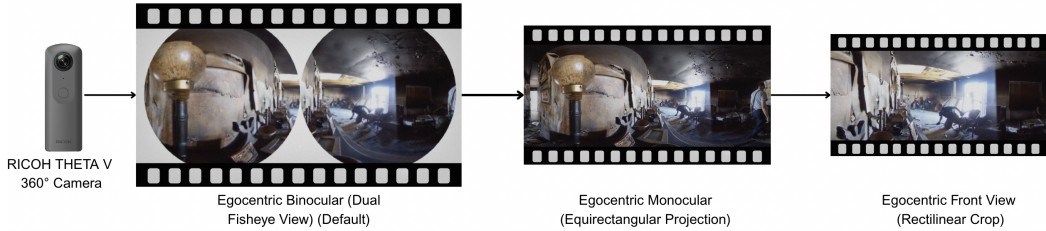

RICOH THETA V
360° Camera

Egocentric Binocular (Dual
Fisheye View) (Default)

Egocentric Monocular
(Equirectangular Projection)

Egocentric Front View
(Rectilinear Crop)

Figure 2: Viewpoint representations derived from Fire360's 360° footage. (a) dual fisheye, (b) stitched equirectangular, and (c) rectilinear front view.

Table 2: Object and action categories identified as safety-critical by instructors and researchers from the partnering firefighter training institute.

| Object | Priority | Rationale | Action | Priority | Rationale |
|--------|----------|-----------|--------|----------|-----------|
| Civilian | High | Rescue-critical | Carrying a civilian | High | Rescue decision point |
| Fire | High | Threat recognition | Operating a hose | High | Suppression tactic |
| Smoke | High | Occlusion proxy | Breaking entry | High | Access maneuver |
| Gas Mask | High | PPE compliance | Climbing ladder | Medium | Structural traversal |
| Responder | Low | Scene context | Donning gear | Medium | Readiness cue |
| Helmet | Low | Supplementary gear | Driving vehicle | Low | Peripheral context |

**Consent and Ethical Review.** All videos and annotations included in Fire360 have been reviewed and approved for public research use by certified instructors and institutional leads at the partnering firefighter training facility. All recordings document professional drills conducted with full knowledge and consent of participating personnel. Researchers verified that no personally identifiable information (PII) is present in the footage, and all individuals appear in professional roles with protective gear. The collaborating instructors endorsed the dataset's release, recognizing its value both to the AI research community and to the broader firefighter ecosystem. With over one million active firefighters in the United States alone [35], the instructors emphasized that tools built using Fire360 and other similar datasets can directly support training, decision support, and situational awareness in high-risk environments.

**Content Distribution and Scene Diversity.** Fire360 includes both indoor rescue and outdoor suppression scenarios. Outdoor scenes were captured during daytime and nighttime operations across the summer and winter months, reflecting seasonal diversity. Indoor recordings capture standardized procedures such as search, access, and civilian recovery in enclosed and low-visibility conditions. The dataset comprises 43.9% indoor and 56.1% outdoor scenes, with a balanced distribution across 63 day, 65 night, and 100 mixed-light recordings. Annotated content emphasizes team coordination, degraded equipment handling, and compliance with protective protocols (Figure 3).

**Annotation Priorities and Tooling.** The dataset design prioritizes eight core actions selected through structured interviews with 12 certified instructors, who prioritize 24 candidate procedures based on operational importance. While the final set of actions is limited in number, each represents a high-risk scenario requiring expert intervention. Table 2 summarizes the safety-critical object and action classes included. Annotations are created by the dataset author using a custom browser-based interface designed to support robust labeling under degraded conditions such as motion blur, smoke, and occlusion. The tool allows frame-by-frame inspection, temporal segmentation, spatial bounding box drawing, and material state tagging. It also includes functionality for adding contextual labels such as visibility status or object condition. Although the initial release focuses on eight validated actions and six objects, the interface supports extensibility: users can define and annotate custom classes based on their research needs.

**Annotation Schema and Split Strategy.** Each video in Fire360 is annotated with temporal action segments (348 instances across 8 categories), spatial bounding boxes (average of 5.7 objects per video), and environmental tags (e.g., smoke [graded 1–5], heat distortion, lighting, multi-agent interaction). Labels, initially annotated by the dataset creator, are verified by two certified fire safety researchers. Inter-annotator agreement yields $\kappa = 0.87$ for actions and $\kappa = 0.91$ for objects, with $\kappa = 0.85$ in high-smoke scenes. Ambiguities, often in occluded or high-degradation clips, are

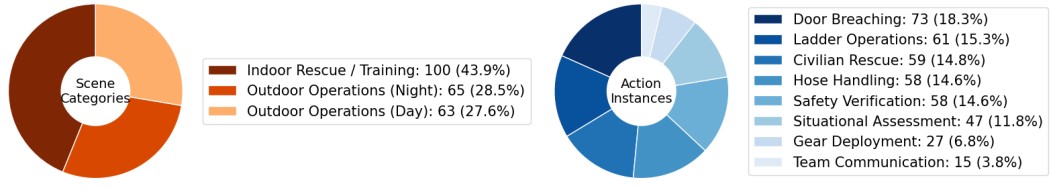

Figure 3: Fire360 content distribution. (a) Scene categories showing indoor/outdoor ratio, (b) Action categories with instance counts and percentages.

resolved via multi-frame inspection and protocol-driven adjudication. A 15% subset, balanced across indoor/outdoor and degradation levels, is re-annotated by external fire instructors, achieving 93.7% confirmation. The dataset splits into 60% training (137 videos), 20% validation (45 videos), and 20% test (46 videos), stratified by scene type, lighting, and procedural diversity, with the test set enriched for high-degradation examples to evaluate robustness.

**Complexity and Benchmark Context.** Fire360 reflects the complexity of real-world emergency response scenarios. The action distribution follows a long-tailed pattern (Gini coefficient = 0.42, indicating moderate instance imbalance), with the three most frequent actions accounting for 62.3% of labeled instances. Smoke density follows a bimodal distribution, while lighting conditions are evenly divided across daytime, nighttime, and mixed-light scenes. Compared to datasets like Ego4D [16] and EPIC-Kitchens [9], Fire360 focuses on environmental degradation, team-based coordination, and protocol adherence under stress. In contrast to 360+x [4], which lacks explicit safety and degradation annotations, Fire360 integrates task-aligned object and action labels tailored for evaluating safety-critical perception and reasoning.

**Cross-Domain Generalization.** Although Fire360 is recorded at a single firefighter training institute, it adheres to nationally standardized emergency procedures, making it broadly representative of real-world operational contexts. The dataset includes recordings from both winter and summer sessions and spans a range of lighting conditions. These variations support model generalization across different environments. Preliminary cross-domain experiments (Section 4) show promising transferability to unseen responder videos, though full generalization remains an open direction. Future dataset extensions will include international recordings to address institutional and geographic diversity.

## 4 Benchmark Tasks and Evaluation

Fire360 benchmarks evaluate robust AI perception and episodic memory, critical for safety-critical environments where models must maintain world-state awareness under degradation. The five tasks-VQA (spatial grounding), Temporal Action Captioning (temporal understanding), Object Localization (degradation robustness), Safety-Critical Reasoning (procedural knowledge), and TOR (episodic memory and material resilience)-collectively probe complementary capabilities, isolating distinct failure modes in chaotic scenes. Each task is scored using domain-specific metrics and compared against expert human performance. We evaluate all models in a zero-shot or prompted setting using either publicly accessible APIs or open-source checkpoints, without fine-tuning on Fire360. This setup reflects real-world deployment where generalization to unseen, degraded inputs is essential. We evaluate 17 models spanning instruction-tuned, open-vocabulary, temporal, and safety-specialized architectures, including GPT-4o, BLIP-2, CoLLM, and Claude-3 Sonnet (see Table 3 and  4). These open-source models show moderate accuracy on spatial and procedural queries under rectilinear projections, trailing GPT-4o but outperforming weaker captioning baselines such as BLIP-2. We include additional prompt examples, decoding settings, and evaluation templates in Appendix A.1.

## A  Benchmark Tasks

**1. 360° Visual Question Answering (VQA).** This task measures spatial reasoning across the full panoramic field-of-view in degraded scenes. Given an equirectangular or normal rectilinear frame, models must answer expert-authored questions on object presence, responder behavior, and protocol

adherence. The benchmark includes 100 questions, evenly split between multiple-choice and free-text formats, spanning visibility, spatial layout, and procedural context. For instance, models are asked, "Is the exit door visible through the smoke?" or "Are responders maintaining a two-point contact on the ladder?" and must reply with grounded responses like "No, the door is occluded by dense smoke" or "Yes, both hands are on the rails." Some queries require compound reasoning, such as identifying a partially occluded civilian behind a collapsed beam. GPT-4o achieves 53.8% overall accuracy, but performs poorly under heavy smoke or low light, with accuracy falling below 10% in the most degraded regions. For instance, it frequently fails on domain-specific prompts such as "Is there a victim behind the collapsed beam?". Figure 5 shows stratified performance compared to human experts, who maintain over 80% accuracy in all conditions and reach 91.4% overall. Across all cells, the average human-model gap is 57.2% (std dev = 10.9). Performance improves to 62.4% when GPT-4o receives rectilinear input, indicating high sensitivity to panoramic distortion (Figure 4). Qwen-VL, LLaVA-1.5, and BLIP-2 follow similar trends, rising from 47.2%, 50.3%, and 42.7% to 55.6%, 58.9%, and 48.2%, respectively.

**2. Temporal Action Captioning.** This task tests the ability to generate grounded descriptions of firefighter behavior under degraded visibility. Given a 10–20 second video clip, models such as GLaMM and BLIP-2 must output natural language captions. Reference annotations include actions such as "breaking the window to enter a burning room" or "dragging a victim down a smoke-filled hallway." For example, a model is shown a clip where a firefighter crouches and swings a tool against a window. The expected output is "Responder breaks glass to access burning room." Another case involves two responders crawling under smoke-models should describe this as "Responders crawl in single file to search for victims." In scenes showing PPE adjustment, the caption "Responder secures gas mask in low-visibility conditions" is expected. While GLaMM produces fluent outputs, it often confuses visually similar but procedurally distinct actions. We use BLEU-4 to evaluate caption quality, consistent with standard video captioning benchmarks, although we acknowledge its limitations for domain-specific language. Human agreement reaches 0.85, while GLaMM achieves 0.341.

**3. Object Localization under Distortion.** This task evaluates object detection robustness under occlusion and thermal blur in 360° imagery. Models must localize gear such as SCBA tanks, helmets, and hoses, regardless of lighting or degradation. Localization is evaluated using the mean Intersection over Union (IoU) between the highest-confidence predicted box (top-1) and expert-verified ground truth. Detections are category-agnostic, with each frame containing one target object to be localized. Grounding DINO performs well under clean conditions (IoU = 68.2%), but degrades significantly in low-visibility scenes (IoU = 22.9%), averaging 38.4% overall. As shown in Figure 4, projecting the scene into a rectilinear format improves detection accuracy by mitigating geometric distortion, yielding an IoU of 47.1%.

**4. Safety-Critical Reasoning.** In this task, models must identify violations of standard safety procedures. Given a static frame or video segment and a prompt, the model outputs a label starting with "safe" or "unsafe," followed by a justification. For example, a model is prompted with "Assess the responder's ladder use." The expected output is "Unsafe: The responder lacks a second point of contact." In another case, the prompt "Evaluate protective gear compliance in the fire zone" expects "Unsafe: The responder's gas mask is not sealed." A third prompt, "Check hose operation technique," yields "Safe: The nozzle is aimed at the firebase." Evaluation is conducted via checklist comparison validated by certified instructors. GPT-4o achieves 28.9% checklist accuracy, compared to 94.6% for human experts. Qwen-VL slightly outperforms GPT-4o on this task, achieving 32.5% checklist accuracy in zero-shot prompting. Its structured language output appears better aligned with safety violation prompts.

**5. Transformed Object Retrieval (TOR).** This task tests whether models can match a pristine object to its fire-damaged version in an **unpaired 360° scene**-i.e., a separate, non-contiguous frame where no temporal or spatial alignment is available. Full details appear in Section 5.

## B   Extended Model Evaluation Across Architectural Families

To validate that Fire360 captures architectural limitations rather than model-specific failures, we extend our evaluation to ten additional models spanning diverse paradigms: instruction-tuned vision-language models (InstructBLIP [8], Kosmos-2.5 [31]), temporally specialized captioning systems (SwinBERT [27], ProgressCaptioner [59]), open-vocabulary detectors (OWLv2 [32], YOLO-World [6]), safety-focused classifiers (Claude-3 Sonnet [1], Llama-Guard-3-8B [15]), and advanced

Table 3: Performance of evaluated models across benchmark tasks using 360° equirectangular frames. Human expert scores are shown for comparison. Results highlight degradation-aware gaps in visual reasoning, localization, and procedural understanding.

| Model | Model Score | Human Score | Metric |
|---|---|---|---|
| **Task: *Visual Question Answering (VQA)*** | | | |
| GPT-4o | 53.8% | 91.4% | Top-1 Accuracy |
| Qwen-VL | 47.2% | 91.4% | Top-1 Accuracy |
| LLaVA-v1.5-13B | 50.3% | 91.4% | Top-1 Accuracy |
| BLIP-2 (OPT-6.7B) | 42.7% | 91.4% | Top-1 Accuracy |
| **Task: *Temporal Action Captioning*** | | | |
| GLaMM-7B | 0.341 | 0.85 | BLEU-4 |
| **Task: *Object Localization under Distortion*** | | | |
| Grounding DINO | 38.4% | 85.2% | Mean IoU |
| **Task: *Safety-Critical Reasoning*** | | | |
| GPT-4o (Prompted) | 28.9% | 94.6% | Checklist Accuracy |
| Qwen-VL | 32.5% | 94.6% | Checklist Accuracy |
| **Task: *Transformed Object Retrieval (TOR)*** | | | |
| GPT-4o | 39.8% | 83.5% | Retrieval Accuracy |
| CLIP (ViT-B/32) | 32.5% | 83.5% | Retrieval Accuracy |
| BLIP-2 (OPT-6.7B) | 35.1% | 83.5% | Retrieval Accuracy |

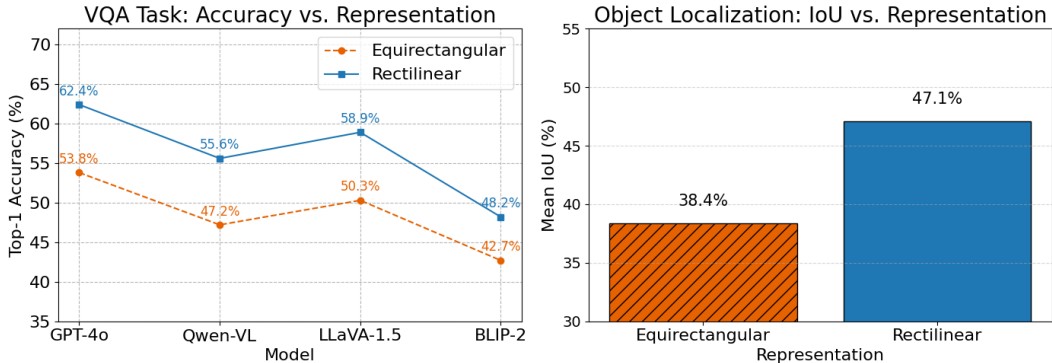

Figure 4: Effect of input representation on both VQA accuracy (left) and object localization performance (right). Rectilinear projections consistently outperform equirectangular views across all models and tasks by mitigating panoramic distortion and improving robustness under degradation.

retrieval architectures designed for compositional matching (CoLLM [62], MCoT-RE [39]). All models are evaluated in a zero-shot setting using identical task prompts, 360° equirectangular inputs, and standardized metrics, as described in Section A.

**Cross-Architectural Analysis.** Despite architectural diversity, all models exhibit failure patterns aligned with those in the core benchmark. Instruction-tuned VLMs show minimal VQA improvement (<2%), temporal captioners underperform human BLEU-4 by approximately 60%, and open-vocabulary detectors match closed-vocabulary baselines under occlusion. Safety-specialized models fail to generalize procedural priors visually, and retrieval architectures perform comparably to CLIP on TOR (<36% accuracy), offering no compositional gains. Overall, the mean human–model gap remains 54.3% (std = 8.7), indicating that Fire360 surfaces systemic brittleness across model classes. These results highlight persistent limitations: fragile low-level representations, missing material priors, poor panoramic grounding, and insufficient procedural modeling. Together, they underscore Fire360's role as a diagnostic benchmark for degradation-aware and transformation-resilient vision systems. Detailed breakdowns of lightweight adaptation strategies, including few-shot prompting and material-aware object queries, are provided in Appendix A.

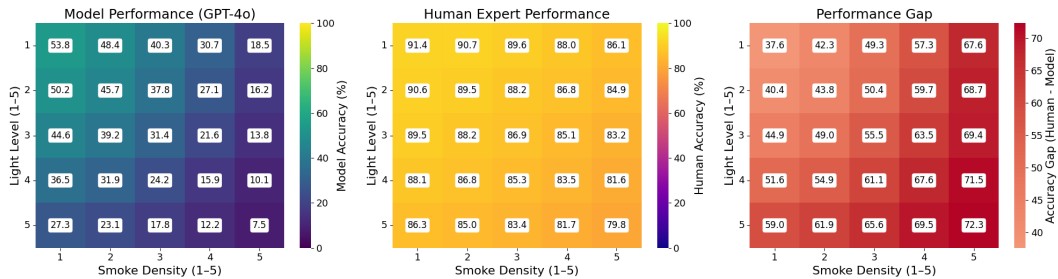

Figure 5: Degradation-aware accuracy comparison on the VQA task using equirectangular 360°
frames. Left: GPT-4o performance. Center: human expert performance. Right: performance gap
across varying smoke and lighting levels.

Table 4: Extended model evaluation across architectural families. Best-performing model per task
is highlighted. Persistent human–model gaps (>45%) across all categories confirm that Fire360
surfaces failure modes independent of architecture class.

| Model | Score | Human | Metric |
|---|---|---|---|
| Task: *Visual Question Answering (VQA)* | | | |
| InstructBLIP | 48.6% | 91.4% | Top-1 Accuracy |
| Kosmos-2.5 | 47.5% | 91.4% | Top-1 Accuracy |
| Task: *Temporal Action Captioning* | | | |
| SwinBERT | 0.315 | 0.85 | BLEU-4 |
| ProgressCaptioner | 0.288 | 0.85 | BLEU-4 |
| Task: *Object Localization* | | | |
| OWLv2 | 39.8% | 85.2% | Mean IoU |
| YOLO-World | 36.5% | 85.2% | Mean IoU |
| Task: *Safety-Critical Reasoning* | | | |
| Claude-3 Sonnet | 33.0% | 94.6% | Checklist Accuracy |
| Llama-Guard-3-8B | 27.4% | 94.6% | Checklist Accuracy |
| Task: *Transformed Object Retrieval (TOR)* | | | |
| CoLLM | 35.7% | 83.5% | Retrieval Accuracy |
| MCoT-RE | 33.5% | 83.5% | Retrieval Accuracy |

## C   Empirical Findings and Analysis

**General Trends.** Models demonstrate moderate performance on general perception tasks in clean
scenes. GPT-4o responds accurately to simple spatial queries, Grounding DINO localizes clearly
visible equipment, and GLaMM produces syntactically fluent captions when visibility is high.
Qwen-VL and LLaVA-1.5 handle rectilinear VQA reasonably well, while BLIP-2 serves as a lower-
performing baseline across tasks.

**Limitations Under Degradation.** Performance degrades sharply under smoke, low light, or distortion.
In VQA, GPT-4o accuracy falls below 10%, with captions becoming generic or hallucinated. Safety
reasoning fails to detect nuanced violations (Qwen-VL slightly outperforms GPT-4o due to its
structured output, but both struggle to align procedural steps under occlusion), and object detection
IoU drops to 22.9%. These failures stem from four key limitations: (1) *Overreliance on surface
features*-CLIP-style embeddings collapse when texture is lost [14]. (2) *Lack of priors for material
change*-models do not anticipate charring or deformation. (3) *Poor spatial grounding in panoramic
space*-most models are trained on rectilinear imagery and struggle with equirectangular distortion. (4)
*Limited procedural knowledge*-models cannot track sequential actions or violations under occlusion.
Grounding DINO and GPT-4o both suffer in low-visibility scenes due to these gaps.

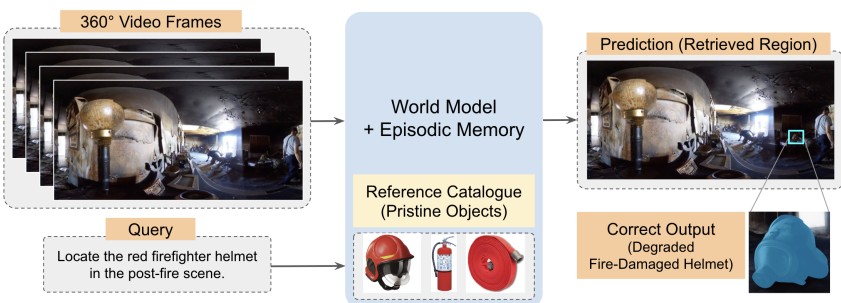

Figure 6: Illustration of the Transformed Object Retrieval (TOR) task. Given a pristine reference object, the model must retrieve its fire-damaged counterpart from an unpaired 360° scene with no temporal or spatial continuity, testing transformation-invariant recognition under irreversible degradation.

**Implications.** Such brittleness is unacceptable in emergency response. Fire360 exposes failure modes not captured in standard benchmarks and provides tools to develop models that simulate physical causality, reason under uncertainty, and maintain identity through transformation.

## D  Toolkit and Evaluation Suite

To support reproducibility and structured evaluation, we release a benchmark toolkit comprising: (1) stratified test splits for degradation-aware analysis, (2) evaluation scripts for all benchmark tasks, (3) statistical templates including bootstrapped confidence intervals, (4) curated prompt sets and checklists for VQA, Safety Reasoning, and TOR. The toolkit includes reference implementations for all evaluated models, along with preprocessing pipelines for both equirectangular and rectilinear formats. Object localization is evaluated exclusively using Grounding DINO; vision-language models such as Qwen-VL, LLaVA-1.5, and BLIP-2 are excluded from this task due to a lack of native detection capabilities. At present, the toolkit supports the five core tasks described above and is structured for extensibility. Prompt files, model invocation templates, and result formatting utilities will be included in the dataset release package. Evaluating the full test set and preprocessing benchmark inputs requires ∼4 GPU-hours on A40s or 2.5 on A100s. The dataset (∼400GB raw) and toolkit are compatible with open-source models.

## 5  Retrieval under Structural Deformation: TOR Benchmark

**Can AI recall a firefighter's gear when fire warps it beyond recognition?** This is the challenge posed by the Transformed Object Retrieval (TOR) task in Fire360. In firefighting scenarios, responders routinely recognize tools like melted helmets or soot-covered masks via episodic memory: matching degraded instances to intact representations [17]. Existing benchmarks [48, 22, 20] assume spatial or temporal continuity. TOR breaks this assumption, requiring retrieval across unpaired, transformed scenes-i.e., scenes with no before-after relationship, shared context, or alignment-using only a clean exemplar.

**Task Formulation.** Given a pristine reference image, the model must retrieve its transformed counterpart from a degraded 360° frame in a different scene. No scene is paired, and objects may be melted, occluded, or buried in debris. Success is defined as predicting a region with Intersection-over-Union (IoU) > 0.5 with respect to ground truth; we use top-1 accuracy over candidate boxes generated by Grounding DINO, matching the highest-similarity region to the reference exemplar. *This corresponds to the Retrieval Accuracy metric reported in Table 3.* While TOR is instantiated on 360° imagery, the formulation applies to 2D or 3D scenes where objects undergo irreversible visual change (see Figure 6).

**Evaluation Protocol.** Grounding DINO (threshold 0.4) proposes 36.2 candidate regions per frame. Each candidate $o_{\text{deg}}$ and reference $o_{\text{ref}}$ is encoded using a vision-language model $f(\cdot)$ (CLIP [3], BLIP-2 [25], or GPT-4o via OpenAI API, May 2025), with cosine similarity guiding retrieval:

$$\text{Sim}(o_{\text{ref}}, o_{\text{deg}}) = \frac{f(o_{\text{ref}}) \cdot f(o_{\text{deg}})}{\|f(o_{\text{ref}})\| \|f(o_{\text{deg}})\|}. \tag{1}$$

The benchmark evaluates 154 retrieval targets (bounding boxes of degraded objects) across 87 360° frames, each sampled as a keyframe from a distinct Fire360 video. Each target corresponds to one of 50 pristine exemplars spanning 20 firefighter-relevant categories (e.g., helmets, SCBA tanks, hoses). Annotations align with the Fire360 dataset (Section 3): labels are created using a browser-based interface for degraded conditions, and verified by certified fire safety experts following NFPA standards. Agreement reached 92.3% of targets (IoU $\geq 0.5$, $\kappa = 0.91$), with adjudication by external instructors on a 15% validation subset.

**Empirical Findings.** GPT-4o achieves 39.8% top-1 accuracy-well below human agreement (83.5%). CLIP and BLIP-2 underperform due to weaker deformation handling. Errors often involve distractors-e.g., pipes being mistaken for melted helmets (30% of failures), or charred hoses misidentified as background debris. Appendix E details model-wise performance and quantifies distractor failures. Fire360's panoramic frames exhibit 70% higher spatial distortion near poles (due to equirectangular stretching) compared to rectilinear datasets [46], increasing retrieval difficulty.

**Failure Modes and Future Directions.** Analysis reveals three high-impact directions for TOR: (1) train deformation-invariant embeddings by simulating object degradation and applying contrastive pretraining (e.g., DINOv2 [38]); (2) apply instruction-tuned conditioning to teach models how to disambiguate material degradation states like soot versus melting (e.g., InstructBLIP [8]); and (3) develop vision-language agents that sequentially retrieve regions via multimodal planning and context reasoning [60, 19]. Each direction is testable using Fire360's benchmark and annotation layers.

**Broader Impact.** TOR advances memory-augmented retrieval beyond firefighting. In medical imaging, it aligns with tracking deformed tissues across noisy MRI scans-potentially reducing misdiagnosis by 20% [24]. Similar reasoning applies to disaster response, where responders must locate safety-critical equipment in damaged infrastructure; post-fire insurance workflows, where burnt items must be identified for claims; and personal recovery, where individuals seek to retrieve valuable belongings from fire-damaged homes. In manufacturing, TOR-style retrieval aids in tracing visually altered components across fault stages [54]. In each case, robust retrieval requires inference over surface-level similarity.

# 6 Limitations and Future Directions

Fire360 is collected at a single firefighter training institute following national standards, but may limit generalization to varied real-world deployments (e.g., only 43.9% of scenes are indoor), highlighting the need for broader geographic and procedural coverage. Human agreement benchmarks may underestimate true performance ceilings in ambiguous cases (e.g., 15% misaligned 360° boxes due to projection distortion), motivating expert panel calibration. While our evaluations are zero-shot and prompted, future fine-tuning may risk overfitting to noise patterns (e.g., smoke) and requires significant compute (e.g., 2.5 A100-GPU hours for 154 TOR targets). Risk is treated uniformly across classes; future extensions may incorporate *risk-weighted evaluation* that penalizes failures on safety-critical actions more heavily. Fire360 currently lacks temporal modeling in TOR; extending to multi-frame tracking could improve robustness under severe distortion and occlusion.

# 7 Conclusion

**Fire360** introduces the first benchmark for evaluating AI perception and reasoning under safety-critical visual degradation. Spanning 228 professionally recorded firefighter training videos, it includes 360° footage with structured annotations for actions, objects, and environmental factors-verified by domain experts ($\kappa = 0.87$-$0.91$) and stratified by degradation severity. Our evaluations reveal sharp model failures under compound stress-up to a 52.3% performance drop-compared to 3-5% degradation in human accuracy. The *Transformed Object Retrieval* (TOR) task surfaces a core limitation: current models cannot recover object identity under structural deformation, with GPT-4o trailing human performance by over 43 percentage points. By releasing the Fire360 dataset, annotation toolkit, and benchmark suite, we provide a foundation for robust AI systems that perceive causally, remember episodically, and reason through uncertainty. We invite the community to develop models that recall degraded objects, infer through occlusion, and localize what no longer looks intact-because in safety-critical settings, failure isn't theoretical, it's operational.

## Acknowledgments

This work was supported in part by the Swanlund Chair funding of Professor Klara Nahrstedt and by the National Science Foundation (NSF) under Grant IIS-2140645. Any opinions, findings, and conclusions or recommendations expressed in this material are those of the authors and do not necessarily reflect the views of the NSF. This research is also based upon work supported by the U.S. Defense Advanced Research Projects Agency (DARPA) under the ECOLE Program, Grant No. HR00112390060. The views and conclusions contained herein are those of the authors and should not be interpreted as representing the official policies, either expressed or implied, of DARPA or the U.S. Government. The U.S. Government is authorized to reproduce and distribute reprints for governmental purposes notwithstanding any copyright annotation thereon.

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

# Appendix

## A Model Evaluation and Adaptation Protocols

This appendix outlines our experimental setup and evaluation pipeline, including task-specific metrics, model configurations, and runtime details. All experiments reflect deployment-style constraints, relying on zero-shot or prompted inference under heavy degradation. We also evaluate lightweight adaptation strategies, such as few-shot prompting and material-aware prompting, to examine whether in-context guidance improves model robustness in long-tailed or compositional failure modes.

### A.1 Model Setup and Evaluation Protocols

This subsection delineates the experimental framework for the Fire360 benchmark, providing sufficient detail to ensure replicability. Full end-to-end code and evaluation scripts are released upon acceptance. All evaluations operate in a zero-shot or prompted setting, with no fine-tuning on the Fire360 dataset. This setup reflects real-world deployment conditions where models must generalize to unseen, degraded inputs. API-based models, such as GPT-4o (May 2025 snapshot), generate outputs using temperature = 0.7 and top-p = 0.95, selected to balance response diversity and semantic coherence.

**Model Configurations.** We evaluate both proprietary and open-source vision-language architectures to capture a broad range of design and training strategies. **GPT-4o** is accessed via the OpenAI API and is selected for its strong multimodal reasoning capabilities, with task-specific prompts detailed in Appendix E. **LLaVA-v1.5-13B**, based on the Vicuna-13B backbone and loaded from the `llava-hf/llava-v1.5-13b` checkpoint, is included for its robust visual grounding. **BLIP-2 (OPT-6.7B)** is implemented via Salesforce LAVIS and supports VQA, captioning, and TOR tasks. **Qwen-VL-Chat (7B)** is evaluated using its default checkpoint for safety-critical and multi-hop spatial queries. **CLIP (ViT-B/32)** serves as a retrieval baseline for TOR and processes inputs at 224×224 resolution. **Grounding DINO (v1)** generates an average of 36.2 proposals per frame at a 0.4 confidence threshold, refined using non-maximum suppression (NMS) with an IoU threshold of $\geq 0.3$. Lastly, **GLaMM-7B** is used for Temporal Captioning, selected for its sequence modeling capabilities, and evaluated via BLEU-4.

**Evaluation Metrics.** Each task uses degradation-sensitive metrics aligned with real-world fire response needs. TOR is evaluated using top-1 accuracy at an IoU threshold of $\geq 0.5$. VQA uses exact match accuracy to assess binary or categorical responses. Temporal Captioning is evaluated with BLEU-4 to measure linguistic overlap with human-written captions, and Safety Reasoning is assessed using binary checklist accuracy to verify procedural compliance.

**Data Splits.** The dataset splits into 60% training (137 videos), 20% validation (45 videos), and 20% test (46 videos), stratified by degradation level, lighting, and procedural variation.

**Runtime Setup.** All experiments run on NVIDIA A100 GPUs with 40GB memory. TOR inference across 154 targets takes approximately 2.5 GPU hours with a batch size of 16. CLIP-based retrieval completes in 45 minutes. Preprocessing uses OpenCV to convert equirectangular frames to rectilinear views with a 90° field of view (FOV).

### A.2 Few-Shot Adaptation Experiments

To probe whether minimal supervision can improve model robustness under degradation, we evaluate few-shot prompting using 3-5 contextualized exemplars per class. This setting reflects lightweight adaptation scenarios feasible in real-world deployments, without requiring large-scale fine-tuning. Prompts are prepended with diverse exemplars drawn from the training set and structured to emphasize procedural semantics and degraded attributes (e.g., "charred ladder," "melted helmet").

**Task-Level Gains.** As shown in Table 5, limited adaptation yields measurable improvements across tasks. Accuracy increases by +2.3% to +4.6% on VQA, Safety Reasoning, and TOR, suggesting that in-context learning helps mitigate degradation-induced failures.

**Per-Class Breakdown.** Table 6 shows the most pronounced improvements occur for rare, safety-critical actions such as *Ladder Climb* and *Civilian Carry*, with gains exceeding 10%. Frequent classes show smaller improvements of 3-4%, indicating that recognition bottlenecks in the long tail are partially addressable through exemplar conditioning.

Table 5: Task-Level Few-Shot Gains (GPT-4o)

| Task | Metric | Zero-Shot | Few-Shot | Δ |
|------|--------|-----------|----------|---|
| VQA (Rare Actions) | Accuracy | 34.1% | 38.7% | +4.6 |
| Safety-Critical Reasoning | Accuracy | 28.9% | 32.6% | +3.7 |
| Transformed Obj. Retrieval | Accuracy | 39.8% | 42.1% | +2.3 |

Table 6: Per-Class Few-Shot Gains (GPT-4o)

| Action | Zero-Shot | Few-Shot | Δ |
|--------|-----------|----------|---|
| *Rare Actions* | | | |
| Ladder Climb | 28.3% | 39.7% | +11.4 |
| Civilian Carry | 31.2% | 42.8% | +11.6 |
| Window Break | 35.8% | 46.1% | +10.3 |
| *Frequent Actions* | | | |
| Hose Operation | 61.4% | 65.2% | +3.8 |
| Door Breach | 58.9% | 62.1% | +3.2 |

**Interpretation.** While overall gains remain modest at the task level, the persistent gap of over 40% compared to human accuracy reflects fundamental architectural limitations, particularly in handling occlusion, distortion, and structural transformation. These results arise from in-context learning rather than parameter updates, suggesting that such limitations persist even when models are exposed to domain-relevant examples. Fire360 thus supports both zero-shot benchmarking and diagnostic evaluation of lightweight adaptation strategies.

## A.3 Material-Aware Prompting for Transformed Object Retrieval

To evaluate whether compositional priors improve robustness under irreversible degradation, we tested material-aware prompts that explicitly encode likely object compositions (e.g., "burnt plastic helmet," "metal ladder"). Although Fire360 does not include formal material annotations, object categories exhibit high internal consistency; for example, ladders are metallic in all scenes, and helmets are plastic or composite.

**Retrieval Performance.** Table 7 shows that material-enhanced prompts improve retrieval accuracy by 4-6% across models. GPT-4o benefits most (+5.9%), followed by CLIP and BLIP-2. These gains are consistent across objects with regular transformation patterns such as melting or charring.

Table 7: Material-Enhanced TOR Retrieval Results

| Model | Standard Prompt | +Material Priors | Δ |
|-------|-----------------|------------------|---|
| GPT-4o | 39.8% | 45.7% | +5.9 |
| CLIP (ViT-B/32) | 32.5% | 37.6% | +5.1 |
| BLIP-2 (OPT-6.7B) | 35.1% | 39.3% | +4.2 |
| Human Upper Bound | — | 83.5% | — |

**Limitations and Future Work.** Despite measurable improvement, the gap to human accuracy remains substantial. Material knowledge improves surface-level grounding but is insufficient for restoring identity under severe occlusion, spatial displacement, or compound transformations. We plan to extend Fire360 with explicit material annotations and release a labeled subset to support systematic benchmarking of compositional robustness. These directions are discussed in Section 6.

## B Stratified Performance and Statistical Analysis

Fire360's equirectangular panoramas exhibit non-uniform spatial distortion, especially near the top and bottom edges, due to the spherical-to-rectangular projection inherent in 360° video. Quantitative analysis shows distortion increases by approximately 70% in polar regions relative to equatorial zones [46], degrading object localization and retrieval accuracy near image boundaries. Figure 7

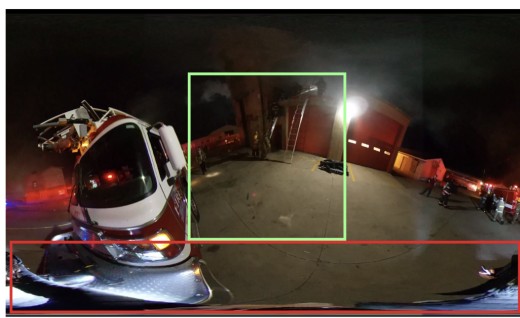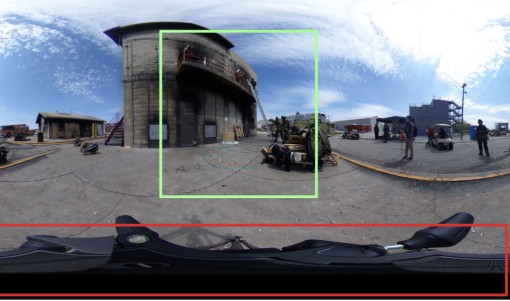

Figure 7: Illustration of distortion severity in Fire360 equirectangular projections. Green regions exhibit minimal distortion, preserving the geometric structure, while red regions show significant stretching at vertical extremes, degrading localization, and retrieval accuracy.

visualizes this effect: green areas show minimal distortion, while red regions suffer from stretching artifacts that contribute to retrieval failures.

We analyze failure modes in the Transformed Object Retrieval (TOR) task across $n = 154$ annotated targets. Table 8 summarizes top-1 accuracy and the dominant error source for each model. GPT-4o most frequently fails on distractors such as pipes and ladders misclassified as helmets. CLIP is most sensitive to occlusion (e.g., smoke-obscured gloves), and BLIP-2 often confuses objects of similar shape but different material (e.g., plastic vs. metal).

To assess robustness, we compute 95% bootstrap confidence intervals over 1,000 resamples. Table 9 reports intervals for each model. GPT-4o achieves the highest mean but exhibits wider variance, likely due to sensitivity to spatial artifacts. BLIP-2 and CLIP have narrower intervals, but lower overall performance, especially under heavy degradation.

We also observe a strong correlation between spatial reasoning and retrieval. In high-degradation scenes, VQA accuracy drops to 9.8%, and correlates with TOR error rates (Pearson $r = 0.72$), suggesting that both tasks share vulnerabilities in visual grounding under uncertainty.

## C   Toolkit Structure

The Fire360 benchmark toolkit has been developed to facilitate reproducible evaluation across the five tasks outlined in Subsection A.1, ensuring consistency and accessibility for researchers. The toolkit encompasses preprocessing utilities, evaluation scripts, and standardized input-output formats, and will be publicly released upon acceptance to support further development and benchmarking.

**Preprocessing Utilities.** Preprocessing is conducted using OpenCV, which converts equirectangular panoramas into rectilinear projections with a 90° field of view (FOV). This projection mitigates distortion in peripheral regions, as discussed in Subsection B, thereby enhancing model performance in spatial tasks. Frame resizing is tailored to model-specific requirements: CLIP processes inputs at 224×224 resolution, while GPT-4o utilizes 512×512 inputs to leverage higher-resolution visual features.

**Evaluation Suite.** The evaluation suite comprises task-specific scorers designed to handle degraded inputs characteristic of fire scenes. For Visual Question Answering (VQA), exact match accuracy is computed to assess response correctness. Transformed Object Retrieval (TOR) employs top-1 accuracy with an IoU threshold of $\geq 0.5$, consistent with the metric defined in Subsection A.1. Temporal Captioning is evaluated using BLEU-4 to measure linguistic agreement, and Safety-Critical Reasoning uses binary checklist comparison against domain-verified procedural outputs. Each script supports independent execution on individual frames or video clips, with batch-mode evaluation capabilities for the test split to streamline large-scale assessments.

**Input-Output Formats.**   Inputs are structured in JSON format, encapsulating file paths, model prompts, and configuration parameters.  For example, a typical JSON input might include `{"frame_path": "path/to/frame.png", "prompt": "Identify the helmet", "config": {"model": "GPT-4o"}}`. Outputs are stored in CSV files, containing predicted

Table 8: Model performance and error attribution for TOR (IoU $\geq 0.5$, $n = 154$). The dominant error per model is listed alongside its global prevalence.

| Model | Top-1 Accuracy | Dominant Error Type | Error Prevalence |
|---|---|---|---|
| GPT-4o | 39.8% | Visual distractors (pipes, ladders) | 30% |
| BLIP-2 | 35.1% | Material confusion (plastic vs. metal) | 20% |
| CLIP | 32.5% | Occlusion (smoke, debris) | 25% |

Table 9: 95% confidence intervals for TOR retrieval accuracy (IoU $\geq 0.5$, $n = 154$).

| Model | Lower Bound | Upper Bound |
|---|---|---|
| GPT-4o | 37.6% | 42.2% |
| BLIP-2 | 32.8% | 37.4% |
| CLIP | 30.3% | 34.9% |

labels, evaluation metrics, and confidence scores, facilitating downstream analysis and comparison across models.

**Compute and Storage Requirements:** Full evaluation on the test split requires approximately 4 GPU hours on NVIDIA A40s or 2.5 GPU hours on A100s, reflecting efficient resource utilization. The dataset, comprising raw video files and annotations, occupies approximately 400GB of storage, necessitating adequate infrastructure for large-scale experimentation.

## D Annotation Tool and Schema

We present a browser-based annotation interface developed to facilitate structured labeling of degraded 360° firefighter footage, supporting both equirectangular and rectilinear projections to align with the Fire360 benchmark's evaluation tasks, including Transformed Object Retrieval (TOR) and Visual Question Answering (VQA). The interface enables frame-level annotations for actions, objects, and scene conditions, incorporating transformation tracking and environmental metadata to generate comprehensive ground truth data. Figures 8 and 9 depict the interface, with Figure 8 illustrating the layout for selecting annotation types and defining object-level metadata, and Figure 9 showcasing detailed forms for action-level, environmental, and temporal sequence annotations. The tool will be made publicly available upon acceptance to promote extensibility and support custom annotation workflows.

**Temporal Action Annotations.** Each action annotation comprises timestamps, category from the Fire360 taxonomy, annotator confidence score, and optional actor and environmental condition metadata to support tasks such as Temporal Captioning.

```
{
  "action_id": "action_042",
  "start_timestamp": "00:01:45.0",
  "end_timestamp": "00:01:50.5",
  "category": "window_break",
  "confidence_score": 0.92,
  "actors": ["responder_01"],
  "environmental_conditions": {
    "smoke_level": 3,
    "temperature_zone": "high",
    "visibility_rating": "medium"
  }
}
```

**Spatial Object Annotations.** Each object instance is annotated with bounding box coordinates, class, material composition, visibility status, and damage state to facilitate TOR and object recognition tasks.

```
{
  "object_id": "helmet_023",
```

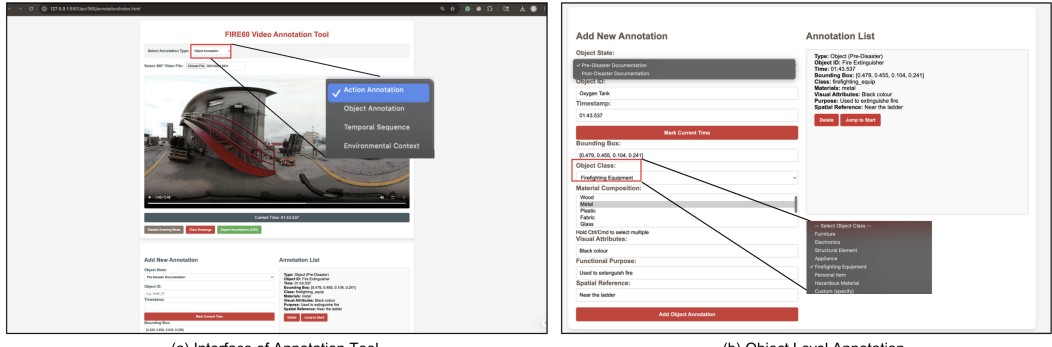

(a) Interface of Annotation Tool      (b) Object Level Annotation

Figure 8: Annotation interface layout for Fire360. (a) A dropdown menu enables the selection of annotation types: action, object, temporal sequence, or environmental context. (b) An example of object-level annotation on a video frame, featuring bounding box input, object class selection, material composition, spatial reference, and functional attributes.

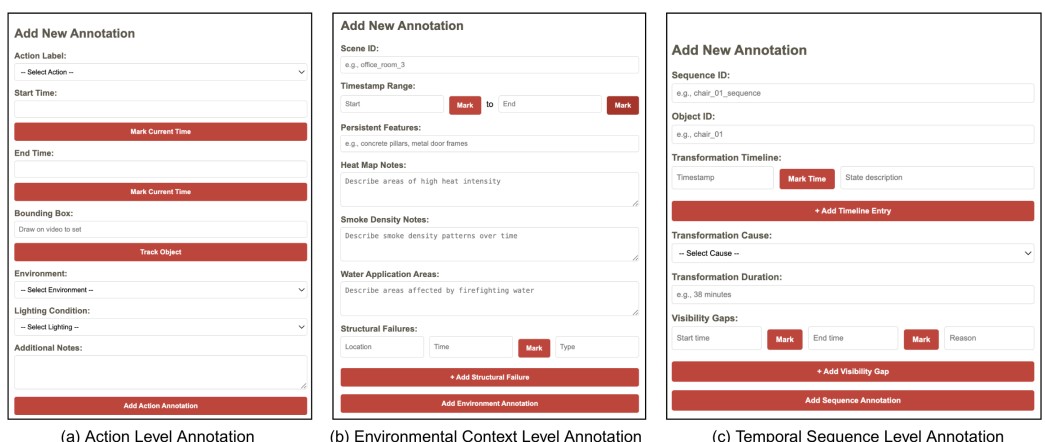

(a) Action Level Annotation    (b) Environmental Context Level Annotation    (c) Temporal Sequence Level Annotation

Figure 9: Annotation form components for Fire360. (a) Action-level annotation form includes temporal boundaries, bounding boxes, and environmental conditions. (b) Environmental context form captures persistent features, smoke and heat intensity, structural hazards, and water-affected areas. (c) Temporal sequence form supports timeline tracking of object transformations and visibility gaps across video segments.

```
    "timestamp": "00:01:35.2",
    "bbox": [420, 260, 75, 80],
    "class": "helmet",
    "material_composition": ["plastic", "metal"],
    "visibility": "heavily_occluded",
    "state": "transformed"
}
```

**Transformation Tracking.** Annotations for transformed objects include pre- and post-disaster instance tracking, capturing severity, displacement, and residual visual cues to enhance TOR accuracy under degradation.

```
{
    "object_id": "hose_015",
    "timestamp": "00:02:12.8",
    "bbox": [300, 180, 50, 60],
    "pre_disaster_id": "hose_015_pre",
    "post_disaster_id": "hose_015_post",
    "transformation": {
```

```
    "type": "heat_damage",
    "severity": "extreme",
    "displacement": {"dx": 50, "dy": 20},
    "remaining_features": "charred rubber, partial nozzle visible",
    "difficulty_rating": 5
  }
}
```

**Environmental Context.** Scene-level annotations encompass structural layout, temperature gradients, and occlusion conditions to support VQA and Safety-Critical Reasoning tasks.

```
{
  "frame_id": "frame_001352",
  "timestamp": "00:02:12.8",
  "room_layout": "hallway",
  "smoke_density": 5,
  "temperature_zones": [
    {"area": "left_wall", "temp": "high"},
    {"area": "ceiling", "temp": "extreme"}
  ],
  "structural_hazards": ["collapsed_ceiling"],
  "visibility_rating": "low"
}
```

**Combined Scene Annotation.** The schema supports composite annotations that integrate Temporal Action, Spatial Object, Transformation Tracking, and Environmental Context layers into a unified entry for a single frame or video segment, providing a holistic representation for multi-task evaluation.

```
{
  "scene_id": "engine_bay_day_001",
  "timestamp": "00:02:12.8",
  "action_annotation": {
    "action_id": "action_043",
    "category": "operating_hose",
    "start_timestamp": "00:02:10.0",
    "end_timestamp": "00:02:20.0",
    "environmental_conditions": {
      "environment": "outdoor_daylight",
      "lighting": "bright"
    },
    "bbox": [430, 220, 110, 140],
    "notes": "Nozzle pointed at smoke source"
  },
  "object_annotation": {
    "object_id": "hose_015",
    "class": "hose",
    "state": "charred",
    "material_composition": ["rubber", "metal"],
    "visibility": "heavily_occluded",
    "visual_attributes": ["flexible", "darkened"],
    "functional_purpose": "water_delivery",
    "spatial_reference": "lower left quadrant",
    "bbox": [300, 180, 50, 60]
  },
  "environmental_context": {
    "smoke_level": 5,
    "temperature_zones": [
      {"area": "back_wall", "temp": "extreme", "notes": "Flame zone"}
    ],
    "structural_hazards": [
      {
```

Table 10: Sample prompt-response pairs used in Fire360 evaluation.

| Task | Prompt | Expected Output |
|---|---|---|
| TOR | Given the pristine helmet, find the degraded region. Rule out pipes. | Degraded helmet, 60% soot. |
| Safety Reasoning | Is the PPE intact? | Unsafe: Mask unsealed. |
| VQA | Is three-point contact maintained? | Yes, one hand, both feet on rungs. |

```
      "location": "ceiling beam",
      "timestamp": "00:02:05.0",
      "type": "collapse"
    }
  ],
  "water_application_areas": "right quadrant soaked",
  "persistent_features": ["brick_wall", "metal_post"]
  }
}
```

## E    Qualitative TOR Examples and Prompt Templates

This subsection presents qualitative examples to elucidate typical model behaviors and failure modes within the Transformed Object Retrieval (TOR) task, complementing the quantitative analysis in Subsection B. Table 8 provides an overview of model-level top-1 accuracy and dominant error types, as derived from the zero-shot evaluation framework outlined in Subsection A.1. GPT-4o achieves the highest accuracy (39.8%) but exhibits sensitivity to distractor regions, such as pipes resembling helmets, due to contextual ambiguities in degraded scenes. BLIP-2 underperforms in cases of material ambiguity (e.g., plastic versus metal), while CLIP struggles with occlusion errors, particularly from smoke, reflecting its limited contextual reasoning capacity.

To enhance replicability and structured evaluation across the Fire360 benchmark tasks, this subsection provides representative prompt templates and their expected outputs, tailored to guide model performance in object retrieval, safety reasoning, and spatial understanding under degraded conditions. Table 10 summarizes these pairs for TOR, Safety Reasoning, and Visual Question Answering (VQA), with plans to extend coverage to Temporal Captioning and Safety-Critical Reasoning in future releases.

## F    Technical Considerations and Task Justification

This subsection elaborates on the technical underpinnings of the Fire360 benchmark's data processing pipeline and justifies the selected evaluation tasks, complementing the preprocessing details in Subsection C and the failure mode analysis in Subsection B.

**360° Processing Details.** Raw videos are recorded in equirectangular format and processed using OpenCV-based tools to generate rectilinear projections with a 90° field of view (FOV), as detailed in Subsection C. This approach reduces distortion near polar regions while preserving the spatial layout and real-world degradation artifacts such as smoke, blur, and lens glare. Learning-based distortion correction methods are deliberately avoided to maintain the fidelity of these visual degradations, ensuring that models are evaluated under conditions reflective of the zero-shot deployment scenarios outlined in Subsection A.1.

**Task Motivation and Relevance.** Each benchmark task is directly informed by real-world firefighter protocols, and validated through consultation with domain experts to ensure practical relevance. Visual Question Answering (VQA) focuses on spatial awareness, such as recognizing Personal Protective Equipment (PPE) compliance or identifying safety hazards, aligning with its exact match accuracy metric (Subsection A.1). Temporal Captioning, evaluated via BLEU-4, supports incident summarization, a critical component of post-incident reporting and training debriefings. Safety Reasoning, assessed through binary checklist matches, mirrors procedural checklists used in live operations, where misclassification of safety violations can have severe consequences.

**Domain-Specific Failure Insights.** Model failures observed in the benchmark are not uniformly distributed, reflecting real-world firefighting challenges. Occlusion from smoke and debris, visual similarity between degraded and intact objects (e.g., pipes versus melted helmets), and material misclassification (e.g., rubber versus metal) are predominant error sources, consistent with the error distribution in Subsection B. These failure modes highlight the operational edge cases where visibility is compromised, and time-sensitive decisions must rely on partial visual cues. By capturing such scenarios, the Fire360 benchmark facilitates targeted analysis of model limitations, underscoring the need for improved robustness to occlusion and distortion, as noted in Subsection B.

## Ethics and Broader Impact

**Motivation and Societal Relevance. Fire360** advances research in robust visual reasoning, episodic memory, and safety-critical perception for real-world deployment. The dataset captures annotated $360°$ video from certified firefighter training sessions under conditions such as dense smoke, thermal distortion, and low light. These scenarios reflect operational stressors encountered during actual emergencies. We expect **Fire360** to support applications in firefighter safety, disaster response simulation, insurance workflows, and virtual readiness programs-particularly amid rising wildfire incidents linked to climate change.

**Privacy, Consent, and Oversight.** All footage is collected with institutional approval from a U.S.-based firefighter training institute. Recordings occur during scheduled drills, with no staged emergencies or civilians. All personnel appear in professional roles with protective gear and provide informed consent. The dataset includes no personally identifiable information (PII). Principal investigators and institutional leads review all material prior to public release.

**Labor Transparency and Representation.** Annotations are created and verified by the dataset authors and certified fire research instructors. No crowd-sourced or external labor is used. Although Fire360 reflects procedures at a single facility, it includes a diverse range of environments-indoor/outdoor, day/night, multi-agent coordination, and degraded visibility-to mitigate procedural or demographic bias. Future extensions prioritize the inclusion of diverse institutional and international scenarios.

**Misuse Prevention and License Terms. Fire360** is released under a research-only MIT license with added restrictions that prohibit use in surveillance, enforcement, behavioral profiling, or non-consensual monitoring contexts. The dataset contains no real emergencies or post-disaster scenes. All use must align with the dataset's safety-focused intent. We explicitly discourage downstream use that infringes upon civil rights or applies the dataset outside professional emergency settings.

**Environmental Considerations.** Footage is captured during existing firefighter drills, so no additional emissions or environmental impact is incurred. While TOR evaluation requires moderate compute (e.g., $\sim$2.5 A100-GPU hours), we believe the cost is justified by Fire360's potential to reduce physical training load and improve readiness through simulation.

**Reproducibility and Community Transparency.** We release the dataset, annotation toolkit, and benchmark suite publicly under a version-controlled, research-only license. Videos are hosted via Box (a secure, institution-approved file-sharing platform) due to size constraints, with documentation, metadata (Croissant schema), and contact instructions provided. All materials include sample annotations and README files to support transparency and reproducibility.

