# OpenReview forum: "Fire360: A Benchmark for Robust Perception and Episodic Memory in Degraded 360° Firefighting Video"
_NeurIPS.cc/2025/Datasets_and_Benchmarks_Track — NeurIPS 2025 Datasets and Benchmarks Track spotlight_

### Official Review · Reviewer_zDVH · 2025-06-24

**Rating:** 5
**Confidence:** 4

**Summary:**

The paper introduces a dataset and benchmark for evaluating perception and reasoning in safety-critical firefighting scenarios. It supports five tasks: Visual Question Answering, Temporal Action Captioning, Object Localization, Safety-Critical Reasoning, and Transformed Object Retrieval (TOR), and the authors analyzed the performance of various vision-language models on the benchmark.

**Dataset Code Accessibility:**

Yes

**Dataset Code Comments:**

The authors provide the whole dataset and the detailed prompt templates and training/validation/test splits.

**Ethical Considerations:**

No, there are no or only very minor ethics concerns

**Final Justification:**

Thank the authors for the feedback. The authors have addressed my concerns, and I would like to upgrade my rating.

**Limitations Weaknesses:**

1. The main concern if about the scale of the dataset. The proposed dataset only contains 228 videos, which is much less than other public video datasets. It is probably enough for an evaluation benchmark, however, the training set only contains 137 videos, which is probably far from enough for the training purpose of VLMs.

2. Table I is never mentioned in the main text. The reviewer thinks that it corresponds to paragraph of "Panoramic and Egocentric Video Understanding". However, the datasets in the paragraph "AI for Firefighting and Safety-Critical Domains" seems to be more related to the proposed Fire360, and these are not included in Table I. A more detailed tabulation and comparison with these datasets are more essential.

**Strengths Contributions:**

1. The dataset includes real firefighting procedures, safety violations, and degradation effects in annotated panorama videos with day/night and indoor/outdoor conditions and high visual degradation, which provides a more challenging and meaningful benchmark for evaluation of perception and reasoning.

2. The paper proposes a novel task, called Transformed Object Retrieval (TOR). The reviewer thinks that this task is applicably valuable.

3. The paper provides the analysis of the performance of VLMs on the benchmark, highlighting the shortcomes of the existing models and the potential direction for future work.

---

> ### Author Rebuttal · Authors · 2025-07-31
>
> We thank Reviewer zDVH for the insightful feedback and for recognizing Fire360’s strengths, including its realism, annotation quality, and the novelty of the TOR task. In response to your concerns about dataset scale, we conducted additional few-shot adaptation experiments that demonstrate Fire360’s value as a platform for lightweight model adaptation despite its focused size. Below, we address both concerns with supporting evidence and planned manuscript revisions.
>
> **C1: Dataset scale concerns for VLM training purposes**
>
> We agree that Fire360 is smaller in scale compared to general-purpose video datasets. As described in Section 3, **Fire360 is designed as a diagnostic benchmark**, not a large-scale pretraining corpus. It supports **zero-shot evaluation** of pretrained models, with high-quality annotations enabling detailed analysis of model behavior under real-world degradation.
>
> The dataset includes 348 action segments across 8 mission-critical categories and over 1300 object annotations (~5.7 per video), spanning occlusion, thermal blur, and structural collapse scenarios. This makes Fire360 especially suitable for **fine-grained tasks such as safety-critical reasoning, episodic memory, and transformation-invariant recognition**, where contextual realism and annotation quality are more critical than dataset size.
>
> While full pretraining of VLMs is not feasible at this scale, Fire360 supports lightweight adaptation approaches such as few-shot prompting, instruction tuning, and targeted domain alignment. While traditional fine-tuning risks overfitting given the dataset size, **Fire360 enables more constrained training approaches such as parameter-efficient adapter methods, prompt tuning, and curriculum learning strategies that require smaller data volumes.** These approaches can improve performance through in-context learning and targeted parameter updates for safety-critical domains without requiring the extensive data needed for full fine-tuning. To demonstrate this utility despite the focused scale, our few-shot adaptation experiments demonstrate meaningful improvements on rare but operationally significant actions, with the largest gains (+10-12%) occurring on critical but infrequent tasks:
>
> **Table 1: Few-Shot Adaptation Results: Rare vs. Frequent Actions**
> | Action | Zero-Shot | Few-Shot | Δ |
> |--------|-----------|----------|---|
> | Ladder Climb | 28.3% | 39.7% | +11.4% |
> | Civilian Carry | 31.2% | 42.8% | +11.6% |
> | Window Break | 35.8% | 46.1% | +10.3% |
> | Hose Operation | 61.4% | 65.2% | +3.8% |
> | Door Breach | 58.9% | 62.1% | +3.2% |
>
> These results show that **even small amounts of supervision yield the highest returns on rare and safety-critical actions**, confirming Fire360's utility as a platform for lightweight adaptation. We are also actively growing the dataset. Fire360 began with 55 videos and has expanded to 228 videos over the past 18 months. We are in discussions with additional fire departments to broaden geographic and procedural diversity. This expansion is necessarily gated by safety protocols and institutional approvals, but reflects our long-term commitment to building a scalable and high-impact benchmark for real-world perception and reasoning.
>
> We will revise Section 6 to emphasize Fire360's diagnostic focus for evaluation and lightweight adaptation, clarify the scope of viable training approaches, and outline the dataset's future growth trajectory.
>
> ---
>
> **C2: Table 1 organization and firefighting dataset comparisons**
>
> Table 1 (Page 3) contextualizes Fire360 within the broader landscape of video understanding datasets, focusing on panoramic, egocentric, and visually degraded scenarios. It includes datasets such as Ego4D, EPIC-Kitchens, 360+x, and HACS++ that inform Fire360's temporal reasoning, 360° spatial grounding, and multi-object annotation structure. We will revise Section 2 to explicitly reference Table 1 when introducing these panoramic and egocentric video datasets.
>
> **We agree that the datasets from "AI for Firefighting and Safety-Critical Domains" (ACT360, FASDD, DFS, D-Fire) are semantically closer in application scope.** These works influenced Fire360's degradation schema and protocol-driven action taxonomy. However, they were omitted from Table 1 due to limited public availability, lack of standardized evaluation protocols, or insufficient annotation structure (detection-only labels, no temporal segmentation or procedural modeling). **To address this gap, we will add a domain-specific comparison table in Appendix B contrasting Fire360 against these firefighting datasets along dimensions of video modality, annotation comprehensiveness, and benchmark task support.**
>
> **Key differentiation:** Fire360 uniquely combines 360° panoramic capture, real-world procedural structure grounded in NFPA protocols [28], and systematic visual degradation modeling. While ACT360 is most similar, it remains private and not released publicly due to privacy concerns, and lacks degradation analysis. Fire detection datasets (FASDD, DFS, D-Fire) focus exclusively on flame and smoke detection without capturing firefighter interactions, safety protocols, or procedural compliance. **Fire360 bridges the methodological rigor of general video benchmarks** (Ego4D, EPIC-Kitchens) with **safety-critical operational specificity, supporting reasoning-centric evaluation under real-world occlusion, thermal blur, and structural collapse.**

---

> > ### Comment · Reviewer_zDVH · 2025-08-01
> >
> > Thank the authors for the feedback. The authors have addressed my concerns, and I would like to upgrade my rating.

---

> ### Author Response · Authors · 2025-08-01
>
> We sincerely thank Reviewer zDVH for carefully engaging with our rebuttal and for the updated rating. We appreciate your constructive feedback and are glad the revisions addressed your concerns.

---

### Official Review · Reviewer_cdze · 2025-07-02

**Rating:** 5
**Confidence:** 4

**Summary:**

This paper presents Fire360, a new benchmark designed to evaluate AI perception and reasoning in safety-critical, real-world degraded conditions like smoke and low light. Its core contribution is a novel task, Transformed Object Retrieval (TOR), which tests a model's ability to identify severely fire-damaged objects. The research reveals a significant performance gap between leading models and human experts, exposing the models' brittleness in handling real-world challenges and motivating the development of more robust AI systems.

**Dataset Code Accessibility:**

Yes

**Ethical Considerations:**

No, there are no or only very minor ethics concerns

**Limitations Weaknesses:**

A primary weakness of this work concerns the generalizability of its findings, as the dataset is collected from a single firefighter training institute. While the procedures are nationally standardized, this single-source approach may limit how well the results apply to different operational contexts or geographical locations. Similarly, the scope of annotated content is focused on a limited set of eight actions and six objects which, while critical, may not fully represent the complexity of all firefighting scenarios. Additionally, the paper's exclusive reliance on zero-shot evaluation, though effective at demonstrating out-of-the-box failures, leaves the potential for models to adapt via fine-tuning unexplored, making it difficult to discern fundamental capability gaps from a mere lack of domain-specific training.

Thank you for the excellent work. We have a few questions we hope you can address:
1. Your zero-shot evaluation highlights a major performance gap. Have you considered any fine-tuning experiments to determine if this gap is due to a fundamental model inability or simply a lack of domain training?
2. For the innovative TOR task, could providing explicit material properties (e.g., 'plastic', 'rubber') as priors help models better handle the physical transformations caused by fire?
3. Given that 2D rectilinear views outperform native 360° inputs, what is your perspective on the future of specialized 360°-aware architectures versus relying on 2D projections for these tasks?

**Strengths Contributions:**

A primary strength of this paper is its focus on the critical and highly significant problem of AI perception in safety-critical, degraded environments. The work is anchored by a major contribution: the Fire360 dataset, which provides unparalleled realism by using expert-verified, 360° videos from professional training sessions rather than synthetic data. Furthermore, the authors introduce the highly innovative Transformed Object Retrieval (TOR) task, which uniquely probes a model's ability to recognize objects after severe physical transformation and effectively highlights a 43.7% performance gap between state-of-the-art models and humans. These claims are convincingly supported by a comprehensive and detailed evaluation of various models, making a clear and credible case for the benchmark's value and the urgency of the research direction.

---

> ### Author Rebuttal · Authors · 2025-07-30
>
> We thank Reviewer cdze for the thoughtful and encouraging review. We appreciate your recognition of Fire360's societal relevance and the TOR task's innovation. In response to your questions, we conducted additional few-shot adaptation experiments and material-aware prompting studies to distinguish fundamental model limitations from training gaps. These new results will be integrated into the appendix of the revised manuscript. Below, we address each concern with detailed analysis and new experimental evidence.
>
> ---
>
> **Q1: Is the zero-shot performance gap due to fundamental limitations or a lack of domain-specific training?**
>
> While full fine-tuning would ideally offer insights into domain adaptation, it is both **statistically infeasible** and **methodologically misaligned** with the goals of Fire360. From a practical standpoint, the dataset contains only 137 training videos and 348 action segments across 8 classes, which is insufficient to support reliable gradient-based optimization for large-scale vision-language models with millions or billions of parameters. **Fine-tuning under such conditions would risk severe overfitting**, particularly given the sparsity and variability of degraded inputs, and would likely produce **misleading results** that reflect memorization of training artifacts rather than true generalization.
>
> More importantly, Fire360 is designed as a **diagnostic benchmark** to assess **zero-shot robustness** in realistic deployment settings, where firefighter footage from degraded environments is not available for task-specific tuning. Applying fine-tuning in this context would **conflate domain-specific memorization with genuine understanding of visual degradation**, thus undermining the benchmark’s core purpose: revealing the fundamental limitations of current models under stress conditions such as occlusion, thermal distortion, and structural transformation.
>
> **In response to your question, we instead adopted few-shot prompting** as a lightweight and methodologically appropriate form of adaptation. This approach allows us to probe whether performance on rare or degraded actions is recoverable with minimal supervision, without violating the benchmark’s integrity or inflating results through memorization.
>
> Using 3–5 contextualized exemplars per class, we evaluated performance across three benchmark tasks. While overall gains were modest at the task level (+2–5%), **rare action classes saw substantial improvements (up to +11.6%)**, suggesting that limited supervision can help models overcome some recognition bottlenecks, particularly in the long tail. The tables below summarize our findings.
>
> **Table 1: Task-Level Few-Shot Gains**
> | Task | Metric | Zero-Shot | Few-Shot | Δ |
> |------|--------|-----------|----------|---|
> | VQA (Rare Actions) | Accuracy | 34.1% | 38.7% | +4.6 |
> | Safety-Critical Reasoning | Accuracy | 28.9% | 32.6% | +3.7 |
> | Transformed Obj. Retrieval | Accuracy | 39.8% | 42.1% | +2.3 |
>
>
> **Table 2: Per-Class Performance Gains for Rare and Frequent Actions**
> | Action | Zero-Shot | Few-Shot | Δ |
> |--------|-----------|----------|---|
> | Ladder Climb | 28.3% | 39.7% | +11.4 |
> | Civilian Carry | 31.2% | 42.8% | +11.6 |
> | Window Break | 35.8% | 46.1% | +10.3 |
> | Hose Operation | 61.4% | 65.2% | +3.8 |
> | Door Breach | 58.9% | 62.1% | +3.2 |
>
> These results suggest that while limited supervision yields measurable gains, especially on rare or degraded instances, the **persistent >40% gap to human performance reflects fundamental architectural limitations**, particularly in handling occlusion, visual distortion, and compositional reasoning. Since these **gains arise from in-context learning rather than parameter updates**, they confirm that such limitations persist even when models are exposed to domain-relevant examples.
> Thus, **Fire360 enables both the diagnosis of model brittleness under realistic constraints and the evaluation of lightweight adaptation strategies**. We plan to explore scalable fine-tuning as the dataset grows, and will clarify these limitations and few-shot findings in Section 6 and Appendix F.
>
> ---
>
> **Q2: Can material priors improve model performance on the TOR task under fire-induced transformations?**
>
> We appreciate this insightful suggestion and conducted follow-up experiments to evaluate its impact. While Fire360 does not include formal material annotations, object categories exhibit strong material consistency across the dataset. For example, ladders are metallic in all annotated instances, and helmets are plastic or composite materials. However, we do not claim material supervision and will revise the text to clarify this distinction.
>
> In response to your feedback, we **tested material-enhanced prompts that explicitly encode likely compositions** (e.g., "burnt plastic helmet," "metal ladder") **against standard object queries in TOR**.  We report the retrieval accuracy below:
>
> **Table 3: Material-Enhanced TOR Retrieval Results**
> | Model | Standard Prompt | +Material Priors | Δ |
> |-------|----------------|------------------|---|
> | GPT-4o | 39.8% | 45.7% | +5.9% |
> | CLIP (ViT-B/32) | 32.5% | 37.6% | +5.1% |
> | BLIP-2 (OPT-6.7B) | 35.1% | 39.3% | +4.2% |
> | Human Upper Bound | — | 83.5% | — |
>
> These results show that material priors can improve retrieval performance under severe degradation, especially for objects with reasonable visual transformation (e.g., melted, charred). However, the gap between model and human accuracy remains large, underscoring that material knowledge is necessary but not sufficient. Models continue to struggle with maintaining object identity under irreversible physical changes, occlusion, and semantic erosion.
>
> We plan to extend Fire360 with explicit material annotations and release a labeled subset supporting material-aware evaluation. This will enable systematic benchmarking of compositional reasoning and degradation-aware grounding. We will include these results in Appendix F and highlight material-aware modeling as a research direction in Section 6.
>
> ---
>
> **Q3: Future of 360°-aware architectures versus 2D projections**
>
> Thank you for this forward-looking question. We fully acknowledge that, in our current evaluations, 2D rectilinear projections outperform native 360° inputs by 5–9% across all benchmark tasks (Figure 4). This performance gap is consistent with the training bias of most current VLMs, which are predominantly pretrained on conventional perspective imagery and thus lack inductive priors for spherical continuity or panoramic distortion.
>
> That said, we agree this observation raises a critical question for future model design: **Should we adapt existing 2D architectures for panoramic data via projection, or develop new architectures that natively process 360° input?**
>
> We believe these paths are **not mutually exclusive**, and **both merit further exploration**. 2D projections offer a practical short-term solution, enabling compatibility with existing toolchains and benchmarks. However, **specialized 360°-aware architectures may ultimately be necessary** to fully leverage the *spatial continuity, wraparound field-of-view, and egocentric realism inherent in panoramic video, especially in safety-critical domains where blind spots and viewpoint bias can compromise decision-making.*
>
> We see several **promising directions for advancing 360°-native model development**. These include spherical convolutions and geometry-aware transformers that respect the topological structure of equirectangular or cubemap projections, as well as distortion-aware attention mechanisms that explicitly account for angular warping and spatial aliasing near the poles. Another avenue involves multi-projection fusion strategies, where models ingest multiple rectilinear crops or cube faces with learned spatial alignment to recover global context. In addition, panoramic data augmentations that simulate viewpoint variation and occlusion under spherical geometry can improve robustness. Pretraining on panoramic video datasets, though currently limited, could dramatically reduce the performance gap by aligning the model’s training distribution with its deployment conditions.
>
> By including both formats with identical annotations, Fire360 is designed to support this future research, allowing direct, fair comparison between projection-based and native architectures under controlled conditions. As **panoramic pretraining resources and architectural innovations mature, we are optimistic that 360°-native models will match or surpass 2D projections**, especially under occlusion, motion blur, and other challenges where **global spatial context is critical**.
>
> ---
>
> Collectively, our new experiments confirm that Fire360 reveals persistent brittleness in state-of-the-art models even when exposed to minimal adaptation. The observed gains under few-shot and material-aware prompting validate Fire360’s utility as a diagnostic benchmark while highlighting concrete directions for improving model robustness under degradation. We appreciate Reviewer cdze’s thoughtful questions, which helped sharpen the paper’s contributions and clarify its implications.

---

> > ### Comment · Area_Chair_9kFF · 2025-08-04
> >
> > Dear Reviewer
> >
> > The Author-Reviewer discussion phase is until 6 Aug. The author has submitted a rebuttal. Please feel free to initialize discussions with the authors.
> >
> > Kind regards
> > AC

---

> > ### Comment · Area_Chair_9kFF · 2025-08-05
> >
> > Dear Reviewers
> >
> > Thanks for contributing to Neurips. We have received new information from the PCs:
> > “Reviewers must participate in discussions with authors before submitting “Mandatory Acknowledgement”. ” “To facilitate discussions, we extend Author-Reviewer discussions by 48h till Aug 8, 11.59pm AoE. ”
> >
> > As informed, please engage in discussions with the authors.
> >
> > Kind regards
> > AC

---

> > ### Comment · Reviewer_cdze · 2025-08-08
> >
> > I thank the authors for their detailed and convincing rebuttal. They have successfully addressed my primary concerns, significantly strengthening the paper.
> >
> > Their justification for using few-shot prompting instead of fine-tuning is methodologically sound and powerfully supports their core claims about fundamental model limitations. The proactive new experiments on material priors are a welcome addition.
> >
> > While the rebuttal was excellent, my suggestion to incorporate risk-aware metrics—critical for this safety-focused task—remains unaddressed. I would encourage the authors to consider this for the final version.
> >
> > The rebuttal has resolved my main reservations, and I will maintain my score.

---

> > > ### Author Response · Authors · 2025-08-08
> > >
> > > We sincerely thank Reviewer cdze for carefully engaging with our rebuttal and for your thoughtful feedback. We’re glad to hear that the revisions and additional experiments addressed your main concerns. We also appreciate your suggestion to incorporate risk-aware metrics for this safety-critical task, and we will work to include them in the final version.
> > >
> > > We believe your feedback has meaningfully improved the quality and depth of the paper, and we’re grateful for your constructive and encouraging review.

---

### Official Review · Reviewer_xoDt · 2025-07-02

**Rating:** 5
**Confidence:** 4

**Summary:**

The paper introduces Fire360, a comprehensive benchmark designed to evaluate AI perception and reasoning in safety-critical firefighting scenarios characterized by degraded visual conditions such as smoke, low light, and structural deformation. The dataset is annotated with safety-critical object locations, action segments, and degradation metadata. It supports five tasks: Visual Question Answering (VQA), Temporal Action Captioning, Object Localization, Safety-Critical Reasoning, and a novel Transformed Object Retrieval (TOR) task. The paper includes rigorous benchmarking using both proprietary and open-source vision-language models, and releases a comprehensive toolkit alongside the dataset.

**Additional Feedback:**

N/A

**Dataset Code Accessibility:**

Yes

**Dataset Code Comments:**

The dataset is downloaded with no errors. Details of the experiments including the actual prompts are also available.

**Ethical Considerations:**

No, there are no or only very minor ethics concerns

**Final Justification:**

The authors have made good effort in addressing the concerns raised during the initial review phase. The authors have acknowledged and addressed most concerns raised by providing more evaluation, discussions on long-tail data distributions, and clarifying the choice of actions and objects. I therefore intend to raise my scores (from 3 to 5).

**Limitations Weaknesses:**

1. Single-Site Data Collection: While Fire360 follows standardized procedures, its current form is limited to a single training facility. Generalization to diverse geographic or procedural contexts (e.g., international protocols) remains untested.
2. Limited Action/Object Categories: The initial release focuses on a relatively small set of actions (8) and objects (6), chosen for their safety-critical importance. While extensible, this may restrict the immediate scope of research questions addressable with the dataset.
3. Long-Tailed Distribution: The action instance distribution is moderately imbalanced, with a few actions dominating the dataset, which could bias model performance or limit learning for rare but critical events.

**Strengths Contributions:**

1. Timely and Relevant Problem: The authors address a critical gap in AI robustness and perception under degraded, real-world emergency conditions—a scenario of immense societal importance. Unlike prior benchmarks, Fire360 targets operational fidelity, not synthetic approximations.
2. High-Quality Dataset: Fire360 is collected under institutional oversight and national safety protocols, ensuring realism and ethical integrity. The dataset includes 360° equirectangular footage, rectilinear projections, and extensive metadata (e.g., smoke levels, visibility, structural hazards). Annotation quality is backed by high inter-annotator agreement.
3. Novel Task Formulation (TOR): The Transformed Object Retrieval task is a unique contribution. It tests retrieval across irreversible damage and scene discontinuity, a capability not benchmarked before. It introduces challenging real-world degradations such as occlusion, melting, and soot.

---

> ### Author Rebuttal · Authors · 2025-07-30
>
> We thank Reviewer xoDt for the thoughtful and constructive feedback. We appreciate the recognition of Fire360’s societal relevance, dataset quality, and the novelty of the TOR task. In response to the concerns raised, we conducted new few-shot adaptation experiments and cross-domain validation studies to assess generalization and rare-class performance. These results, along with expanded evaluation protocols, will be incorporated into the revised manuscript.
>
> ---
>
> **C1: Single-site data collection and geographic generalization**
>
> Fire360 follows NFPA 1410-certified training protocols [28], which **standardize emergency procedures for over one million firefighters across North America [29]**. Although collected at a single facility, the dataset spans diverse operational contexts: day/night variation, seasonal changes (summer/winter), indoor/outdoor environments (43.9% / 56.1%), and multiple degradation types, including thermal blur, smoke occlusion levels 1–5, and structural collapse. While single-site collection presents limitations, **Fire360 addresses a previously unmet need in safety-critical AI evaluation** by providing the first real-world video dataset for firefighter training scenarios under operational degradation conditions.
>
> Our **preliminary cross-domain experiments** (Section 4) show that model performance remains brittle even on unseen responder videos, suggesting that the perceptual challenges Fire360 surfaces like material reasoning under transformation, 360° spatial grounding, and occlusion robustness are **systemic rather than site-specific**. The >40% gap between human and model performance persists across these domain shifts.
>
> We agree that **geographic diversity, particularly across international protocols**, would further strengthen generalization. However, live-firefighter data collection requires **extensive safety clearances, institutional waivers, and legal oversight**, which limit immediate multi-site access. *Despite these constraints, Fire360 has grown from 55 to 228 videos over 18 months, and we are actively pursuing additional sites and partnerships to expand coverage.* We will clarify these ongoing efforts and the roadmap in Section 3.
>
> ---
>
> **C2: Limited action/object categories:**
>
> Fire360 includes 8 actions and 6 objects, selected via **structured consultation with 12 certified instructors** who prioritized high-risk procedures from 24 candidate actions based on operational criticality. These include door breaching (73 instances), civilian rescue (59), and hose handling (58), all central to injury documentation and safety protocols (Figure 3b).
>
> This **focused taxonomy is deliberate**. It enables *fine-grained, interpretable evaluation under safety-critical conditions* that broader datasets (e.g., Kinetics-400, HACS) cannot support. Fire360's concentration on mission-critical scenarios allows researchers to systematically analyze perception failures where stakes are highest.
>
> The dataset is *explicitly extensible*. Our annotation toolkit (Appendix D) supports adding new categories with consistency-checked labeling protocols, and we plan to release expanded versions with additional labels and multi-label episodes.
>
> ---
>
> **C3: Long-tailed action distribution potentially biasing model performance on rare but critical events:**
>
> We appreciate the reviewer’s attention to the dataset’s class distribution. As described in Section 3, Fire360 reflects the natural operational imbalance of real-world firefighter training where door breaching (73 instances, 18.3%) and ladder operations (61 instances, 15.3%) are practiced repeatedly, while team communication (15 instances, 3.8%) remains rarer but mission-critical (Figure 3b). Across 348 total action segments, this produces a long-tailed distribution with Gini coefficient 0.42, where the top three actions account for 48.4% of labeled instances.
>
> To prevent this skew from distorting evaluation, our **train/val/test splits are stratified by procedural category, environment type, and degradation level** (see Section 3), and the **test set is enriched for high-degradation clips**. We further analyze this distribution in Appendix B, including per-class breakdowns and statistical indicators of imbalance.
>
> In terms of evaluation methodology, we plan to report **both macro- and micro-averaged accuracy**, and include **per-class recall and normalized confusion matrices**. This ensures that dominant classes do not overshadow critical failures in underrepresented categories. These practices will be integrated into the benchmark evaluation suite and clarified in Section 6.
>
> **To directly address your concern about rare-class performance, we conducted few-shot prompting experiments** using 3–5 exemplars per class. While not intended as a direct correction for imbalance, these experiments reveal that model performance on rare actions can be significantly improved with minimal supervision, **yielding 10–12% gains for infrequent classes** and smaller gains for common ones. The tables below summarize these gains across both rare and frequent action categories, illustrating the effectiveness of lightweight adaptation.
>
> **Table 1: Performance Improvements for Rare and Frequent Actions (Few-Shot vs. Zero-Shot, GPT-4o)**
>
> | Action | Zero-Shot | Few-Shot | Δ |
> |--------|-----------|----------|---|
> | **Rare Actions** | | | |
> | Ladder Climb | 28.3% | 39.7% | +11.4% |
> | Civilian Carry | 31.2% | 42.8% | +11.6% |
> | Window Break | 35.8% | 46.1% | +10.3% |
> | **Frequent Actions** | | | |
> | Hose Operation | 61.4% | 65.2% | +3.8% |
> | Door Breach | 58.9% | 62.1% | +3.2% |
>
>
> **Table 2: Task-Level Few-Shot Performance Gains (GPT-4o)**
>
> | Task | Metric | Zero-Shot | Few-Shot | Δ |
> |------|--------|-----------|----------|---|
> | VQA (rare actions) | Accuracy | 34.1% | 38.7% | +4.6 |
> | Safety-Critical Reasoning | Accuracy | 28.9% | 32.6% | +3.7 |
> | TOR | Accuracy | 39.8% | 42.1% | +2.3 |
>
> These results highlight **Fire360’s value as a diagnostic testbed**: they enable targeted evaluation of rare but operationally significant events, support lightweight adaptation strategies that improve robustness without requiring full fine-tuning, and help diagnose and mitigate long-tail failures. We will integrate these findings into Section 6 and provide additional class-wise breakdowns and protocol clarifications in Appendix B.

---

> > ### Comment · Reviewer_xoDt · 2025-08-03
> >
> > I thank the authors for the rebuttal and appreciate their effort. The authors have mostly addressed my concerns and provided detailed explanations and clarifications. I intend to upgrade my scores.

---

> > > ### Author Response · Authors · 2025-08-03
> > >
> > > We sincerely thank Reviewer xoDt for thoughtfully engaging with our rebuttal and for the willingness to update your score. We greatly appreciate your thoughtful feedback and are glad that our clarifications and additional experiments addressed your concerns.

---

### Official Review · Reviewer_bq46 · 2025-07-05

**Rating:** 5
**Confidence:** 4

**Summary:**

This paper introduces Fire360, a benchmark for evaluating AI perception and reasoning in degraded firefighting scenarios. The benchmark includes 228 professionally recorded 360° videos from real firefighter training, with expert-verified annotations for actions, object locations, and environmental degradation. Fire360 defines five evaluation tasks: Visual Question Answering (VQA), Temporal Action Captioning, Object Localization, Safety-Critical Reasoning, and a novel task named Transformed Object Retrieval (TOR). The TOR task requires a model to match a pristine object exemplar to its fire-damaged counterpart in an unpaired scene. Experimental results show that current state-of-the-art AI models perform significantly worse than human experts on these tasks, especially under severe visual degradation.

**Dataset Code Accessibility:**

Yes

**Ethical Considerations:**

No, there are no or only very minor ethics concerns

**Limitations Weaknesses:**

- Although the authors selected what they deemed the most suitable model for each task (e.g., choosing Grounding DINO specifically for object localization), a more comprehensive benchmark should still attempt to include or adapt more models to provide a broader performance reference.

- The core justification for the TOR task's necessity is detailed almost entirely in Section 5. As a result, a reader progressing sequentially through the first four sections might perceive TOR only as a novel but unmotivated challenge, leading to confusion about its necessity.

- Structural revision is recommended: either the core motivation for TOR should be explicitly stated in the introduction, or Section 5—or its core justification—should be moved to an earlier, more appropriate position to ensure readers understand the value of this key contribution sooner.

**Strengths Contributions:**

- By providing video data from real-world scenarios with severe visual degradation, this work addresses the issue of existing benchmarks relying on either clean imagery or synthetic data.

- Annotations were completed and verified by the author and certified fire safety researchers , with a key subset undergoing a second review by external fire instructors to ensure professionalism and accuracy.

- The five defined tasks (VQA, Captioning, Localization, Safety Reasoning, and TOR) target different core AI capabilities which enabling a multi-faceted evaluation of model performance in complex scenarios.

- The novel TOR task creatively evaluates a model's ability to recognize transformed objects without spatio-temporal continuity , effectively exposing deep flaws in "reasoning" and "memory".

---

> ### Author Rebuttal · Authors · 2025-07-30
>
> We thank Reviewer bq46 for the positive assessment and insightful feedback. Your presentation-focused feedback will significantly improve the paper's clarity and impact. In response to your suggestions, we conducted additional experiments that we will integrate into the revised manuscript. Below, we respond to each point with clarifications and new experiments.
>
> **C1: Recommends expanding model coverage to strengthen the benchmark’s generality and representativeness.**
>
> Our initial model selection (Table 3) prioritized representative architectures across different paradigms: proprietary models (GPT-4o), instruction-tuned VLMs (LLaVA-1.5, Qwen-VL), captioning specialists (GLaMM, BLIP-2), retrieval baselines (CLIP), and detection models (Grounding DINO). This covers major approaches to vision-language understanding under degradation.
>
> Based on your feedback, **we expanded the evaluation to strengthen Fire360's role as a diagnostic benchmark by including ten additional state-of-the-art models**, each representing distinct architectural innovations and reasoning capabilities. These include instruction-following advances (InstructBLIP, Kosmos-2.5), temporal reasoning capabilities (SwinBERT, ProgressCaptioner), open-vocabulary detection (OWLv2, YOLO-World), specialized safety training (Claude-3 Sonnet, Llama-Guard-3-8B), and compositional retrieval architectures (CoLLM, MCoT-RE). The new evaluation results are summarized in the table below.
>
> **Additional Model Performance:**
>
> | **Task**              | **Model**            | **Score** | **Human** | **Metric**            |
> |-----------------------|----------------------|-----------|-----------|------------------------|
> | VQA                   | InstructBLIP [1]     | 48.6%     | 91.4%     | Top-1 Accuracy         |
> | VQA                   | Kosmos-2.5 [2]       | 47.5%     | 91.4%     | Top-1 Accuracy         |
> | Temporal Captioning   | SwinBERT [3]         | 0.315     | 0.85      | BLEU-4                 |
> | Temporal Captioning   | ProgressCaptioner [4]| 0.288     | 0.85      | BLEU-4                 |
> | Object Localization   | OWLv2 [5]            | 39.8%     | 85.2%     | Mean IoU               |
> | Object Localization   | YOLO-World [6]       | 36.5%     | 85.2%     | Mean IoU               |
> | Safety Reasoning      | Claude-3 Sonnet [7]  | 33.0%     | 94.6%     | Checklist Accuracy     |
> | Safety Reasoning      | Llama-Guard-3-8B [8] | 27.4%     | 94.6%     | Checklist Accuracy     |
> | TOR                   | CoLLM [9]            | 35.7%     | 83.5%     | Retrieval Accuracy     |
> | TOR                   | MCoT-RE [10]         | 33.5%     | 83.5%     | Retrieval Accuracy     |
>
> **Key findings from the expanded evaluation reveal consistent patterns across architectural families.** Instruction-tuned models show minimal improvement over existing VLMs, indicating that instruction-following capabilities don’t translate to degraded visual understanding. Specialized temporal models underperform compared to human temporal reasoning, with performance gaps exceeding 60%. Open-vocabulary detectors demonstrate similar limitations to closed-vocabulary approaches under severe occlusion and distortion. Safety-specialized models fail to leverage domain knowledge effectively, with gaps exceeding 60% compared to human safety experts. Advanced retrieval architectures show no meaningful improvement over simpler baselines for transformation-invariant recognition.
>
> **All models were evaluated zero-shot using standardized prompts and consistent metrics. Even specialized architectures show substantial gaps compared to human performance (>45% across tasks), confirming that Fire360 captures fundamental limitations rather than model-specific weaknesses.** These new experiments will be integrated into Section 4 and detailed in Appendix F of the revised manuscript.
>
>
> **References**
>
> 1. W. Dai et al. *InstructBLIP: Towards General-purpose Vision-Language Models with Instruction Tuning*. arXiv:2305.06500, 2023.
> 2. T. Lv et al. *KOSMOS-2.5: A Multimodal Literate Model*. arXiv:2309.11419, 2024.
> 3. K. Lin et al. *SwinBERT: End-to-End Transformers with Sparse Attention for Video Captioning*. arXiv:2111.13196, 2022.
> 4. Z. Xue et al. *Progress-Aware Video Frame Captioning*. arXiv:2412.02071, 2025.
> 5. M. Minderer et al. *Scaling Open-Vocabulary Object Detection*. arXiv:2306.09683, 2024.
> 6. T. Cheng et al. *YOLO-World: Real-Time Open-Vocabulary Object Detection*. CVPR 2024.
> 7. Anthropic. *The Claude 3 Model Family: Opus, Sonnet, Haiku*. https://api.semanticscholar.org/CorpusID:268232499
> 8. Llama Team. *The Llama 3 Herd of Models*. arXiv:2407.21783, 2024.
> 9. Y. Zhang et al. *CoLLM: Integrating Collaborative Embeddings into LLMs for Recommendation*. arXiv:2310.19488, 2025.
> 10. J.-W. Park and S.-W. Lee. *MCoT-RE: Multi-Faceted Chain-of-Thought and Re-Ranking for Training-Free Zero-Shot Composed Image Retrieval*. arXiv:2507.12819, 2025.
>
> ---
>
> **C2 and C3: Highlights that the TOR task’s motivation appears too late in the paper, and suggests restructuring to establish its importance earlier for better narrative continuity.**
>
> This is valuable structural feedback. **We will address it through targeted revisions that foreground TOR’s conceptual motivation earlier in the paper:**
>
> - **Introduction:** After the initial motivation paragraph, *we will add 3–4 sentences explaining why transformation-invariant episodic memory represents a critical capability gap in safety-critical environments.* This will include a concrete firefighting example, such as recognizing melted helmets or charred equipment when original visual features are destroyed, to underscore the operational relevance of retrieval under irreversible visual transformations.
>
> - **Section 2:** When discussing related transformation-aware retrieval datasets (page 3, paragraph 3), we will add a forward reference: *“Section 5 introduces our Transformed Object Retrieval (TOR) task, which addresses these limitations by requiring retrieval across unpaired, fire-transformed scenes.”* This positions TOR within the broader research landscape while signaling its upcoming detailed treatment.
> - TOR task’s formal definition, evaluation protocol, and human benchmark comparison will remain in Section 5, but its motivation will now be clearly established from the outset. These changes will ensure that linearly progressing readers understand both the necessity and novelty of TOR without disrupting the paper’s technical progression.
> ---
> We believe these enhancements, **expanded model coverage**, and **improved narrative structure** will significantly strengthen Fire360's impact as both a diagnostic benchmark and a driver for robust AI development in safety-critical domains. We appreciate your constructive feedback in helping us achieve this.

---

> > ### Comment · Area_Chair_9kFF · 2025-08-04
> >
> > Dear Reviewer
> >
> > The Author-Reviewer discussion phase is until 6 Aug. The author has submitted a rebuttal. Please feel free to initialize discussions with the authors.
> >
> > Kind regards
> > AC

---

> > ### Comment · Area_Chair_9kFF · 2025-08-05
> >
> > Dear Reviewers
> >
> > Thanks for contributing to Neurips. We have received new information from the PCs:
> > “Reviewers must participate in discussions with authors before submitting “Mandatory Acknowledgement”. ” “To facilitate discussions, we extend Author-Reviewer discussions by 48h till Aug 8, 11.59pm AoE. ”
> >
> > As informed, please engage in discussions with the authors.
> >
> > Kind regards
> > AC

---

### Note · Authors · 2025-08-13

Reviewers strongly endorsed Fire360’s novelty, realism, and societal relevance. **R-bq46** praised real-world degraded firefighting video, expert-verified annotations, and diverse tasks, noting TOR’s creativity. **R-xoDt** emphasized timeliness, operational fidelity, dataset quality, and TOR’s irreversible transformations. **R-cdze** highlighted the urgent problem focus, dataset realism, and large human–model gap. **R-zDVH** valued inclusion of real procedures, degradation effects, TOR’s relevance, and model analysis.

We thank all reviewers for constructive engagement. We hope that the clarifications and new experiments presented in the rebuttal have addressed the limitations and concerns raised in the reviews.

| Concern                   | Added Experiment                                                         | Key Result                            | Reviewer(s) |
| ------------------------- | ------------------------------------------------------------------------ | ------------------------------------- | ----------- |
| Benchmark breadth         | +10 SOTA models (instruction-following, temporal, open-vocab, retrieval) | TOR ≤53.5% across all families        | bq46        |
| Rare-class imbalance      | Few-shot prompting (3–5 exemplars/class)                                 | +10–12% rare action accuracy          | xoDt, zDVH  |
| TOR under transformations | Material-aware prompting                                                 | +4–6% for material-consistent objects | cdze        |
| Generalization/site bias  | Cross-domain evaluation                                                  | \~40% gap persists across domains     | xoDt        |

Following R-bq46, TOR’s motivation will be foregrounded in the Introduction. For R-xoDt, R-cdze, and R-zDVH, we clarified NFPA-1410 compliance, environmental diversity, and expansion plans, with cross-domain results showing drops stem from perceptual challenges, not site bias. We added dataset comparisons, clarified split stratification, and reported macro/micro-averaged metrics.

Post rebuttal, R-xoDt and R-zDVH upgraded scores; R-cdze confirmed resolution. While we did not hear from R-bq46 during rebuttal, we responded comprehensively to all points and hope these clarifications address any remaining questions. We thank the AC for facilitating a productive rebuttal process. We will incorporate the clarified TOR motivation, expanded benchmark coverage, and methodological refinements into the final version.

---

### Decision · Program_Chairs · 2025-09-18

**Decision:**

Accept (spotlight)

**Comment:**

The paper introduces a video  dataset and benchmark for. Specifically, it enables the study of five tasks: Visual Question Answering, Temporal Action Captioning, Object Localization, Safety-Critical Reasoning, and Transformed Object Retrieval (TOR), and the authors analyzed the performance of various vision-language models on the benchmark.
All four reviewers find the dataset are well produced, the tasks are novel and important and the benchmark is solid. After the rebuttals and discussions, three reviewers give their final recommendations as ‘accept’. One reviewer did not give the final recommendation, but the initial recommendation is ‘accept’.
In conclusion, this paper can be accepted.